# Rapid electron transfer by the carbon matrix in natural pyrogenic carbon

Tianran Sun[1], Barnaby D.A. Levin[2], Juan J.L. Guzman[3], Akio Enders[1], David A. Muller[2,4], Largus T. Angenent[3,5,6] & Johannes Lehmann[1,5]

Surface functional groups constitute major electroactive components in pyrogenic carbon. However, the electrochemical properties of pyrogenic carbon matrices and the kinetic preference of functional groups or carbon matrices for electron transfer remain unknown. Here we show that environmentally relevant pyrogenic carbon with average H/C and O/C ratios of less than 0.35 and 0.09 can directly transfer electrons more than three times faster than the charging and discharging cycles of surface functional groups and have a 1.5 V potential range for biogeochemical reactions that invoke electron transfer processes. Surface functional groups contribute to the overall electron flux of pyrogenic carbon to a lesser extent with greater pyrolysis temperature due to lower charging and discharging capacities, although the charging and discharging kinetics remain unchanged. This study could spur the development of a new generation of biogeochemical electron flux models that focus on the bacteria–carbon–mineral conductive network.

[1] Department of Soil and Crop Sciences, School of Integrated Plant Sciences, College of Agriculture and Life Sciences, Cornell University, Ithaca, New York 14853, USA. [2] School of Applied and Engineering Physics, College of Engineering, Cornell University, Ithaca, New York 14853, USA. [3] Department of Biological and Environmental Engineering, College of Agriculture and Life Sciences, Cornell University, Ithaca, New York 14853, USA. [4] Kavli Institute for Nanoscale Science, Cornell University, Ithaca, New York 14853, USA. [5] Atkinson Center for a Sustainable Future, Cornell University, Ithaca, New York 14583, USA. [6] Center for Applied Geosciences, University of Tübingen, Tübingen, 72074, Germany. Correspondence and requests for materials should be addressed to T.S. (email: ts689@cornell.edu).

Electron transfer reactions are the basis of biogeochemical cycles in water, soil, and sediment and govern most geochemical and biochemical transformations[1,2]. Natural organic matter, due to the charging and discharging cycles of its surface functional groups, such as quinone/hydroquinone pairs, has been shown to serve as a 'geobattery' that can reversibly accept and donate electrons[3,4]. A similar geobattery mechanism has also been found in the iron redox cycles in natural magnetic minerals[5]. In addition, organic matter may mediate electron transfers between redox-active compounds in soils and sediments[2,6,7]. Such redox processes have been demonstrated to play important roles in suppression of greenhouse gas emissions[4,8], iron mineral reduction[9] and decontamination[10]. Recently, pyrogenic carbon has been found to reversibly accept and donate large amount of electrons[11], and functions very similarly to natural organic matter in many biogeochemically and environmentally relevant redox reactions, such as reductions in nitrous oxide emission and total denitrification[12], iron mineral reduction[13,14] and organic contaminant transformation[15]. Since pyrogenic carbon is now recognized as a ubiquitous and major component of natural organic matter worldwide[16,17], the contribution of pyrogenic carbon to electron fluxes could be considerable. While surface functional groups, mainly quinones, are known to be associated with redox properties of pyrogenic carbon[11], the surface electrochemistry of pyrogenic carbon matrices has not been investigated. This novel electrochemical behaviour determines direct electron transfer across the interface of pyrogenic carbon matrices and external electron donors and acceptors, and therefore is different from the established electron transfer through pyrogenic carbon matrices due to bulk electrical conductivity[18]. Electrochemical studies provide means to quantify the interface electron transfer kinetics of pyrogenic carbon matrices and reveal the transition of kinetically preferred pathways between carbon matrices and surface functional groups for electron transfer at naturally occurring pyrolysis temperatures. In contrast to the known geobattery mechanism of surface functional groups, we propose to refer to the interface electron transfer by pyrogenic carbon matrices as a 'geoconductor', which directly transfers electrons that are generated or consumed by surface electrochemical reactions elsewhere. Here we have examined the kinetics of electron transfer through environmentally relevant pyrogenic carbon matrices separately from the charging and discharging behaviour of surface functional groups using cyclic voltammetry in which pyrogenic carbon was used as a working electrode and an immobilized redox couple, respectively. Both direct electron transfer, and charging and discharging processes were characterized by the height of the peak current and the potential at which the current was highest as a result of oxidation and reduction reactions. We quantified electron transfer behaviours to mineral phases and related the magnitude and kinetics of electron transfer to the order of carbon structures in pyrogenic carbons.

## Results

**Electrochemical properties of pyrogenic carbon matrices.** Electrochemical tests demonstrated that pyrogenic carbons can transfer electrons by means of direct electron transfer of carbon matrices (Fig. 1a,b) and the charging and discharging cycles of surface functional groups (Fig. 1c,d). The carbon matrices were clearly able to transfer electrons at a rate that exceeded those caused by charging and discharging cycles of surface functional groups (Fig. 1e,f). This fast transfer through the carbon matrices occurred only at pyrolysis temperatures above 600 °C (wood biomass, pyrolysis duration was 30 min) with corresponding molar H/C and O/C ratios of $<0.35$ and 0.09, respectively (the

elemental composition of pyrogenic carbon samples can be found in Supplementary Table 1). A sigmoidal kinetic pattern (Fig. 1e) was obtained with an exponential increase of the direct electron transfer rate constant in the temperature range from 600 to 725 °C, followed by a gradual increase to 800 °C. (dimethylaminomethyl) ferrocene was used to characterize the electron transfer kinetics of pyrogenic carbon matrices due to its fast electron transfer response and a formal potential falling into the middle of the pyrogenic carbon potential window (see the selection of redox couples in the Methods section and Supplementary Fig. 1). Detailed information of rate constant calculations can be found in the Methods section and Supplementary Figs 2 and 3. To provide a theoretical maximum kinetics for pyrogenic carbons, electron transfer by graphite has also been tested and a rate constant of $0.021 \, cm \, s^{-1}$ was obtained (Supplementary Fig. 4), which was only slightly higher than those of the pyrogenic carbons pyrolysed at 800 °C ($0.018 \, cm \, s^{-1}$ with molar H/C and O/C ratios of 0.11 and 0.05). Therefore, further increases in electron transfer kinetics will be marginal with lower H/C and O/C ratios at pyrolysis temperatures exceeding 800 °C.

Due to the inherent inner resistance caused by minerals and non-aromatic carbon structures, it was not possible to precisely calculate the rate constant of pyrogenic carbon between 650 and 725 °C, and the actual values lay in the grey area between estimated upper and lower limits (Fig. 1e). For pyrogenic carbon produced at 400–600 °C with molar H/C and O/C ratios of 0.62–0.35 and 0.24–0.09, however, no peak current but only weak current passage were observed, which indicated that electron transfer through the pyrogenic carbon matrices was very slow (Fig. 1b). We estimated the rate constant at this temperature range by exponential regression of the kinetic curve from higher temperatures. No current flow was detected for pyrogenic carbon pyrolysed at 400 °C with molar H/C and O/C ratios of 0.62 and 0.24, and therefore it was used as a practical zero kinetic point for regression analysis. Surface functional groups dominated the redox properties of pyrogenic carbon at 400–600 °C. The electron transfer kinetics of pyrogenic carbon matrices shown in Fig. 1e were corrected for resistance and conductivity (see the correction of inner resistance in the Methods section), and therefore only show the effects of surface carbon structures instead of bulk conductivity, and are independent of surface functional groups such as quinones.

**Functional group composition.** Reversible charging and discharging current peaks revealed that surface functional groups of pyrogenic carbon can act as both an electron acceptor and donor (Fig. 1c,d). Quinone and hydroquinone functional groups have previously been identified as the most common couple that is responsible for the redox activities of natural organic matter and pyrogenic carbon[11,19,20]. Using attenuated total reflectance Fourier transform infrared (ATR-FTIR) spectroscopy and electron energy loss spectroscopy (EELS), we confirmed the presence of quinone moieties in the studied pyrogenic carbons (Fig. 2). FTIR spectra indicated that significant changes in functional group chemistry of pyrogenic carbon occurred above a pyrolysis temperature of 300 °C. Formation of ketones, aromatic carbon and carboxyl groups[21] were identified by the vibration of C=C and C=O stretches at 1,700 and 1,600 cm$^{-1}$. The increased absorption ratio of aromatic C–H to aromatic C=C bonds suggested the formation of larger and increasingly condensed aromatic sheets. Nearly all featured vibrations disappeared above a temperature of 700 °C indicating attenuated functionalities. Further carbon k-edge EELS examination of the pyrogenic carbons at fine spatial scale revealed similar trends in which diminished surface functionality occurred with increased pyrolysis temperature, as shown by the lower

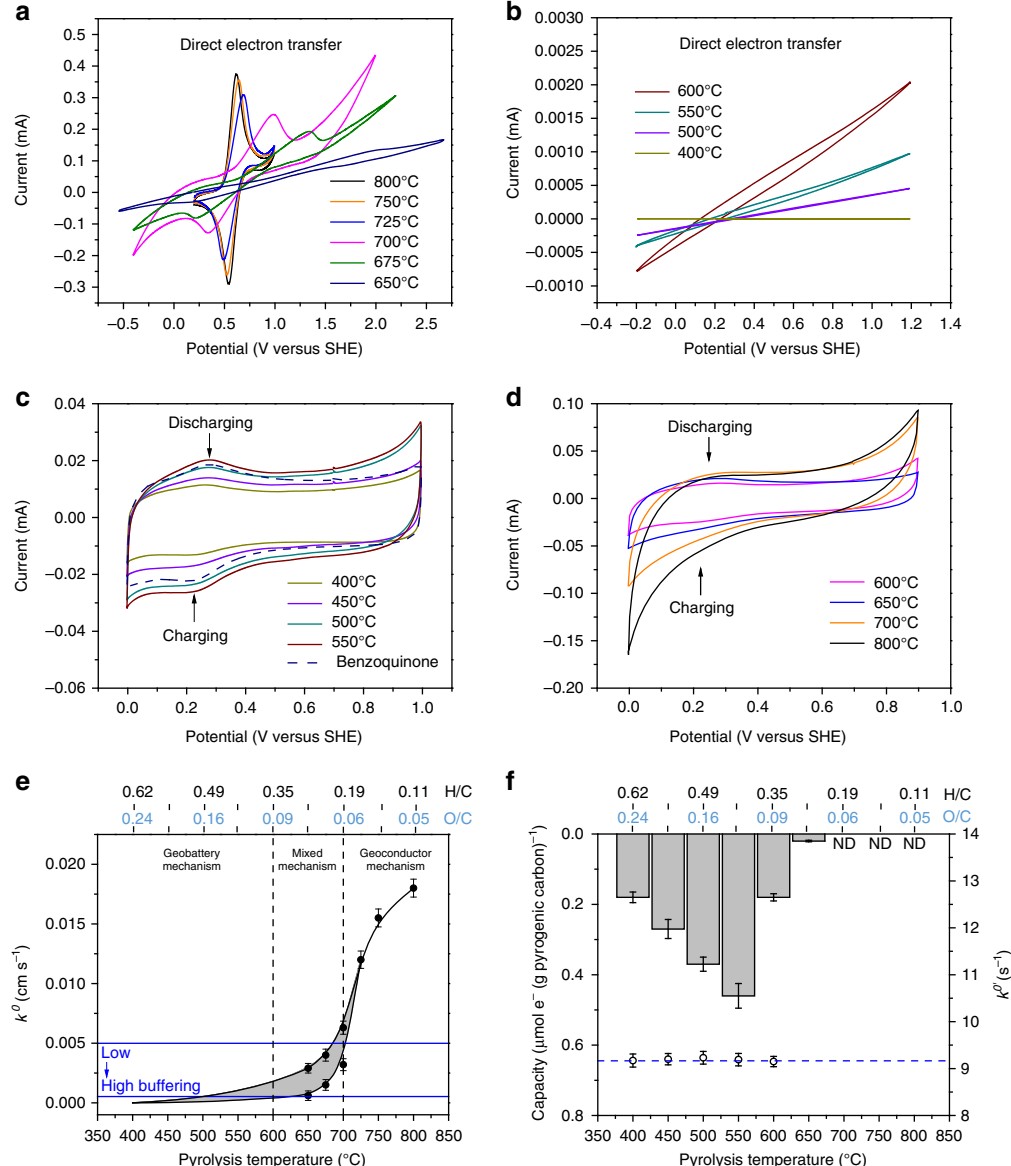

**Figure 1 | Direct electron transfer versus charging and discharging. (a,b)** Direct electron transfer. Cyclic voltammograms of (dimethylaminomethyl) ferrocene (3 mM) in 0.1 M KCl (pH = 7) at the pyrogenic carbon working electrodes. A formal potential of 0.59 V versus standard hydrogen electrode (SHE) was obtained for (dimethylaminomethyl)ferrocene. Scan rate = 150 mV s$^{-1}$ for pyrogenic carbons produced at 800–675 °C and 50 mV s$^{-1}$ for pyrogenic carbon produced at 650–400 °C. **(c,d)** Charging and discharging cycles. Cyclic voltammograms of immobilized surface functional groups (carried by 0.12 mg pyrogenic carbon pyrolysed at temperatures 400–800 °C) at a graphite working electrode. Dashed line indicates the cyclic voltammogram of immobilized benzoquinone (0.03 mg on graphite electrode). Scan rate = 100 mV s$^{-1}$. **(e)** Direct electron transfer rate constant ($k^0$) of pyrogenic carbon matrices. Blue lines are the $k^0$ of benzoquinone measured at low- to high-buffered conditions (pH = 7). **(f)** Estimated charging and discharging capacity (column) and rate constant ($k^{0\prime}$) of surface quinone groups (open circles) and benzoquinone (blue dashed line). ND denotes not detectable. The corresponding molar H/C and O/C ratios are given above the top x axes of **e,f**. Error bars are s.d. of triplicate measurements.

intensity of transitions between 285 and 292 eV. The 1s-π* transition at around 284.1 eV (expressed as shoulders at a lower energy of the C=C transition peak at 284.8 eV) along with the transition peak at 286.1 eV suggest the existence of olefins[22] and heteroaromatic quinone moieties[23] such as benzoquinones[24,25]. These features highly resembled spectral properties of pyrogenic carbon generated from woody biomass[11,21], in which the quinone groups amounted to 3.1–1.2 mmol g$^{-1}$ carbon made at pyrolysis temperatures of 400–700 °C.

**Charging and discharging of surface quinone groups.** The charging and discharging capacities increased from 400 to 550 °C

(columns in Fig. 1f), likely due to the increased reactivity of quinone groups in this intermediate temperature range[11]. The capacities sharply declined above 550 °C with no detectable capacity found above 700 °C, corresponding to the attenuated surface functionality shown in Fig. 2. The measured capacity was 1,000 times lower than previously reported values using so-called mediated chronoamperometry[11,26,27]. This is probably because only a small fraction of the redox-active moieties of the materials were in contact with the working electrode that allowed for electron transfer (Supplementary Fig. 5a and ref. 28). The cyclic voltammetry we used in this study with immobilized pyrogenic carbon as a redox couple is more effective in determining the charging and discharging kinetics (Supplementary Fig. 3) and

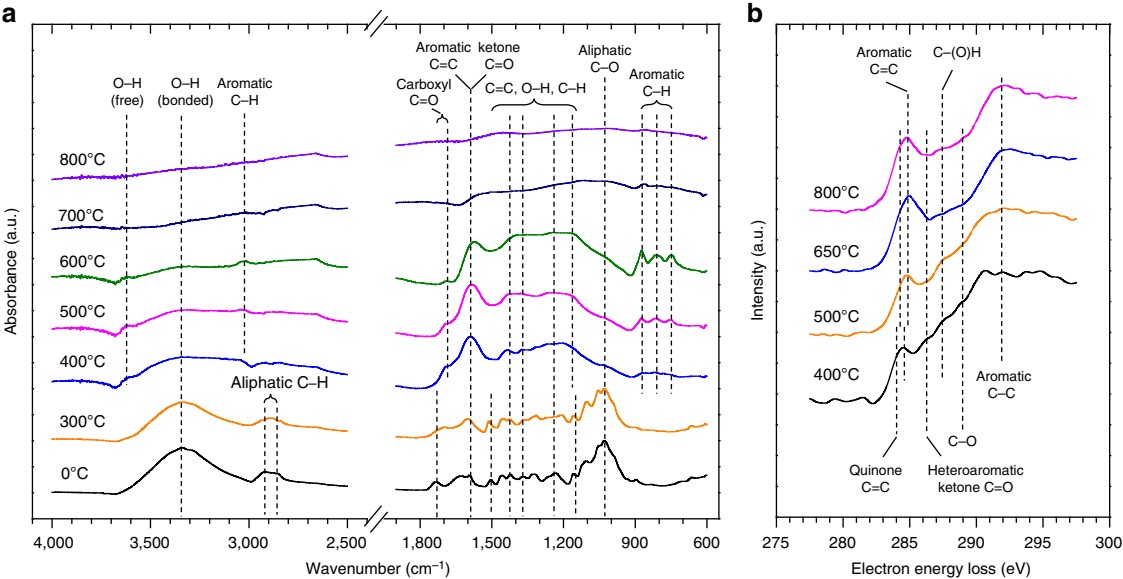

**Figure 2 | Spectroscopic examination of the surface functionality of pyrogenic carbon.** (**a**) Fourier transform infrared (FTIR) spectroscopy of pyrogenic carbon samples pyrolysed at temperatures from 300 to 800 °C, including a non-pyrogenic wood biomass as indicated by 0 °C. (**b**) Electron energy loss spectroscopy (EELS) of pyrogenic carbons obtained at temperatures from 400 to 800 °C. Peaks corresponding to aromatic and quinone C = C bonds shifted to lower energy side on pyrogenic carbon produced at 400 °C than that of pyrogenic carbon produced at greater temperatures. Individual EELS spectra were recorded while scanning the electron beam over about 100 × 100 nm areas of a thin section of the pyrogenic carbons. The data presented are averages of spectra from 5 to 10 different pyrogenic carbon particles at each temperature. These are more representative than spectra from single particles. The Fourier ratio method was used to reduce the effects of multiple electron scattering on the spectra[70].

is independent of the concentration of the surface quinone groups[29]. In contrast to the electron transfer through the pyrogenic carbon matrices, the surface quinone groups possessed unchanged kinetics in charging and discharging with greater pyrolysis temperature (open circles in Fig. 1f). The formal potential of surface quinone groups was linearly and negatively related to solution pH with slopes of − 56 and − 53 mV pH$^{-1}$ for pyrogenic carbon produced at 450 and 550 °C, respectively (Supplementary Fig. 6), which are in agreement with the theoretical Nernstian prediction that the formal potential of quinone compounds shift with pH at a rate of − 59 mV pH$^{-1}$ (for a two-electron and two-proton transfer process). We attributed the discrepancy (between our measurements of up to − 56 mV pH$^{-1}$ and the theoretical value of − 59 mV pH$^{-1}$) to the impaired pH buffering on pyrogenic carbon in the microenvironment protected by Nafion than in a theoretical Nafion-free environment.

**Comparison of electron transfer kinetics**. Pyrogenic carbon in water forms a suspension rather than a solution, therefore the charging and discharging rate of the surface quinone groups that is comparable to the direct electron transfer of pyrogenic carbon matrices cannot be directly obtained by cyclic voltammetry under diffusive conditions. In accordance with the structural similarity as demonstrated by the EELS spectra, we found that surface quinone groups of pyrogenic carbon also behaved electrochemically very similar to the benzo- and hydroquinone couple by matching their redox potential (Fig. 1c and Supplementary Table 2) and kinetic performance (Fig. 1f and Supplementary Fig. 3). Therefore, benzoquinone was used as a model compound for surface quinone groups to compare the charging and discharging rate to the direct electron transfer rate of pyrogenic carbon matrices.

Redox behaviour of quinone compounds can contain several steps and is highly pH dependent. A nine-membered square

scheme is generally used to describe the electron transfer and protonation of quinone compounds across a wide pH range and including both aqueous and aprotic solvents[30]. Under environmental conditions (that is, pH neutral and aqueous soil solution), two major redox couples are involved[31]. In a fully buffered solution, the conversion between quinone and hydroquinone represents a slow charging and discharging cycle coupled with protonation; the interconversion between quinone and the mixture of quinone dianion and hydrogen bonded quinone dianion in an unbuffered solution represents a fast charging and discharging cycle[32]. These two redox couples are likely to co-exist in an intermediately buffered solution. Soil generally has a high diversity[33] in buffering capacities due to different parent materials and pedogenesis. Therefore, we estimated the rate constant of benzoquinone in both unbuffered (0.0051 cm s$^{-1}$) and fully buffered (0.0013 cm s$^{-1}$) solutions (pH = 7 for both solutions, using glassy carbon as a working electrode, see Supplementary Fig. 7) to provide a range that covers a wide variety of soil buffering conditions (blue lines in Fig. 1e).

The estimated rate constants of benzoquinone were in the range of direct electron transfer by carbon matrices produced at 600–700 °C (Fig. 1e). A wider $k^0$ range (0.000152–0.0072 cm s$^{-1}$) was reported[34–36] for anthraquinone-2,6-disulfonate (AQDS, a soil organic matter analogue in extracellular electron transfer[3,4]), which covered direct electron transfer rate of pyrogenic carbon matrices made at 500–700 °C. Therefore, we concluded that the kinetically preferred electron transfer by carbon matrices over the surface quinone groups of both pyrogenic carbon and soil organic matter forms at pyrolysis temperatures higher than 700 °C. Three temperature ranges (Fig. 1e) were identified for the studied pyrogenic carbons from low to high pyrolysis temperature, which correspond to three mechanisms of pyrogenic carbons for electron transfer, namely a geobattery mechanism where charging and discharging of quinone groups dominate (400–600 °C with molar H/C and O/C ratios of 0.62–0.35 and 0.24–0.09), a mixed

geobattery and geoconductor mechanism (600–700 °C with molar H/C and O/C ratios of 0.35–0.19 and 0.09–0.06), and a geoconductor mechanism with kinetically preferred electron transfer (700–800 °C with molar H/C and O/C ratios of 0.19–0.11 and 0.06–0.05).

**Graphitic carbon and electrochemical properties.** We used Raman spectral mapping to investigate how changes in the structural properties of the pyrogenic carbon matrices may influence the rate of direct electron transfer, due to its *in situ* examination of the carbon surfaces through which the electron transfer occurs. Figure 3a shows Raman spectra acquired from pyrogenic carbons with pyrolysis temperatures of 800, 650 and 500 °C. The *G* peak in the Raman spectra (Fig. 3a) arises due to the vibration of carbon atoms in $sp^2$ sites, and the *D* peak arises due to the disorder in aromatic ring structure in $sp^2$ carbon atoms[37]. As pyrolysis temperature increased, the ratios of the *D* to *G* peak intensity in the Raman spectra rose (Fig. 3b), which indicates greater ordering and growth of graphitic structures in the pyrogenic carbon[37]. The trends observed in the *D* to *G* ratios, and in the *D* and *G* peak positions are similar to those observed in carbon films after annealing at different temperatures[38], and in coal at different levels of maturity[39]. Growth of ordered pyrogenic carbon structures was relatively modest between pyrolysis temperatures of 400–600 °C and as a result, direct electron transfer by the carbon matrices in this temperature range

remained slow. As pyrolysis temperature increased above 600 °C and the H/C and O/C ratios dropped below 0.35 and 0.09, the carbon became increasingly graphitic, which was consistent with the observed increase in electron transfer kinetics. Raman spatial maps of the *D* peak and *G* peak intensities did not show any features that were unique to either peak, suggesting that the structure of the pyrogenic carbons was homogeneous at the micron length scale for all pyrolysis temperatures analyzed (Fig. 3c). Raman spectra of pyrogenic carbons produced at all pyrolysis temperatures are available in the Supplementary Fig. 8.

**Metal electron acceptors.** The pyrogenic carbon matrices demonstrated a wide potential range of about 1.5 V in directly transferring electrons to different kinds of acceptors (Fig. 4a). This potential range covers the entire redox potential of thermodynamically favourable redox couples reported in soils and sediments[4,40]. The electrons were transferred by pyrogenic carbon matrices, which was confirmed by the linear increase of the peak current as a function of the increased scan rate (Fig. 4b). Different peak currents in mineral phases indicated different reduction rates controlled by the surface area and reactive site density[41]. As expected, manganese oxides had the most positive reduction potential so that electrons transferred through pyrogenic carbon matrices are first accepted by manganese oxide, followed by iron(III) chloride and iron minerals. Further, the distinction of reduction potentials between surface quinone

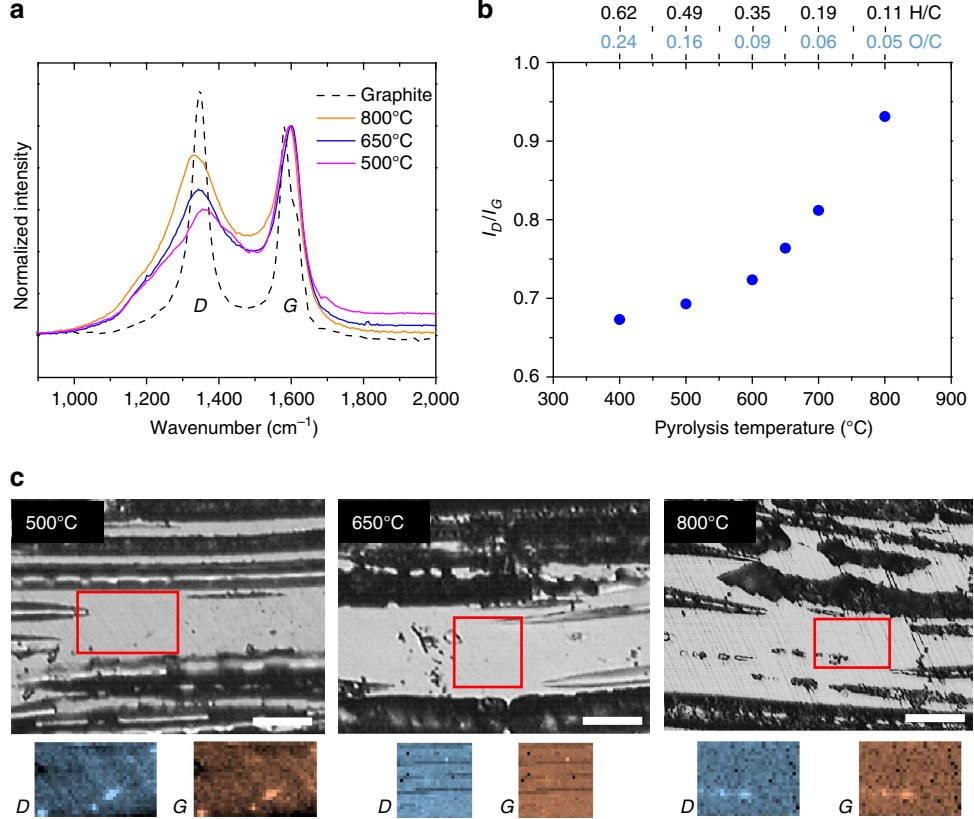

**Figure 3 | Raman spectral mapping.** (**a**) Raman spectra of pyrogenic carbon matrices. Chart legend indicates pyrolysis temperature. Dashed line indicates the spectrum of commercial graphite from Sigma-Aldrich. Each spectrum is a sum of >600 single spectra taken over ∼25 × 25 µm areas of pyrogenic carbons and a graphite surface to provide representative spectra with high signal-to-noise ratio. (**b**) The ratio of *D* to *G* peak intensity at different pyrolysis temperatures. The intensity ratio was calculated based on peak heights obtained by fitting a Lorentzian curve to the *D* peak, and a Breit–Wigner–Fano curve to the *G* peak. The corresponding molar H/C and O/C ratios are given above the top *x* axes of **b**. (**c**) Optical microscopy images of finely polished sections of pyrogenic carbon at different pyrolysis temperatures. Raman spatial maps of *D* and *G* peak intensities were acquired from areas indicated on the optical image. Scale bars are 25 µm. Surface roughness of areas mapped are <2 µm.

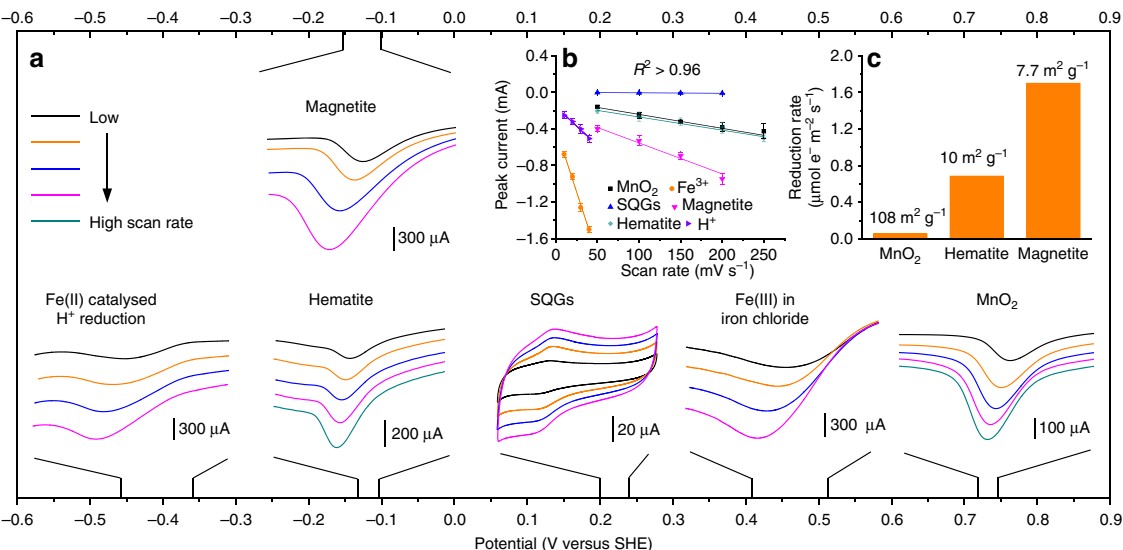

**Figure 4 | Direct electron transfer from pyrogenic carbon matrices to minerals.** (**a**) Linear sweep voltammograms of minerals on pyrogenic carbon. All linear sweep voltammetry were performed by immobilizing minerals on the surface of a pyrogenic carbon (pyrolysed at 800 °C) working electrode. Scan rates varied from 50 to 250 mV s$^{-1}$ with an interval of 50 mV s$^{-1}$, except for iron(III) chloride with scan rates from 10 to 40 mV s$^{-1}$ at an interval of 10 mV s$^{-1}$. Fe(II) catalysed H$^+$ reduction followed Fe(III) reduction in iron chloride at more negative potential, and was not separately prepared. The cyclic voltammetric chart shows the direct electron transfer by pyrogenic carbon matrices to surface quinone groups (SQGs, measured using 0.4 mg pyrogenic carbon produced at 500 °C). The actual range of peak potentials from low to high scan rates are shown at the main *x* axis for all tested species. (**b**) The linear relationships between scan rate and peak current of all tested species. Error bars are s.d. of triplicate measurements. (**c**) The reduction rate comparison among minerals. Numbers on the top of columns indicate the Brunauer–Emmett–Teller surface area for each mineral.

groups and metal minerals strongly suggested that the metals contained in the pyrogenic carbon were not responsible for electron transfer and that the detected charging and discharging property of pyrogenic carbon (here in Fig. 4 represented by pyrogenic carbon produced at 500 °C) was therefore mainly a result of quinone moieties[11]. Fe(III) in iron chloride was reduced more rapidly than those in the mineral phases and showed two continuous current peaks in one voltammogram: a Fe(III)/Fe(II) reduction peak (at about +470 mV versus SHE) followed by a second peak at more negative potential (at about −400 mV versus SHE) which we attributed to the Fe(II) catalysed H$^+$ reduction. Reduction of H$^+$ ions did not appear during the reduction of mineral phases when we scanned the potential to the same negative value. These two peaks were manually broken into two separate peaks shown in Fig. 4 for better demonstration of iron mineral reductions as their reduction potentials lay in between. Magnetite and hematite minerals showed a very similar reduction potential that ranged from −150 to −100 mV versus SHE from a high to low scan rate, which was consistent with previously reported potentials (about −200 to −120 mV versus SHE) obtained by thermodynamic calculations and microbial mineral reduction[41–43]. The observed reduction rate (the rate was normalized by the mineral surface area, Fig. 4c) of magnetite was higher than those of hematite and manganese oxides, in spite of the fact that the reduction rate might have been affected by the specific adsorption of phosphate (used as a pH buffer) on mineral surfaces. Apart from the direct electron transfer to external acceptors such as minerals, electrons can also be stored in the graphene-like sheet structures in pyrogenic carbon matrices[44] and then be rapidly released when the electron acceptors become accessible (that is, the geocapacitor mechanism)[45]. Our results (Supplementary Fig. 9) demonstrated that the electrochemical capacitance of pyrogenic carbon increased with an increase in pyrolysis temperature. The kinetics of the electron storage and release process of the geocapacitor mechanism is expected to be closer to those of the geoconductor mechanism as they share

carbon matrices as the electron transfer pathway instead of through the charging and discharging pathway of surface quinone groups in its geobattery mechanism.

## Discussion

This study investigated the interface electron transfer of pyrogenic carbon from a kinetic point of view, and demonstrated the transition from a mediated electron transfer exclusively by quinone groups to a direct electron transfer dominated by the carbon matrices as a result of greater graphitic structures with lower O/C and H/C ratios caused by an increase of pyrolysis temperature. Amorphous carbon structures generated at low temperatures caused both a high inner resistance and an energy barrier that limited electron transfers through carbon matrices to the surface of low-temperature pyrogenic carbons. Therefore, redox cycles by the geobattery mechanism in such amorphous carbons dominated the electron flow of pyrogenic carbon (Pathway 1 in Fig. 5). Graphitic carbon structures that were able to conduct significant electron transfers emerged at intermediate temperatures. In the temperature range of 600–700 °C and molar H/C and O/C ratios of 0.35–0.19 and 0.09–0.06, respectively, similar kinetic performance was obtained between the geoconductor and geobattery mechanism. However, considering the decreased capacities of the geobattery, the geoconductor may dominate the actual electron flow and 'shortcut' the geobattery by direct electron transfer from external donors to acceptors (Pathway 2). More ordered carbon structures with H/C and O/C ratios below 0.19 and 0.06 in high-temperature pyrogenic carbons above 700 °C created a rapid pathway that conducted electron transfers more than three times faster than the redox cycles of the geobattery mechanism (Pathway 3). We predict a gradual reduction of direct electron transfer by pyrogenic carbon matrices during natural ageing processes (Supplementary Figs 10–12) that may lead to extensive breaking of carbon double bonds especially for carbon matrices produced at lower temperature.

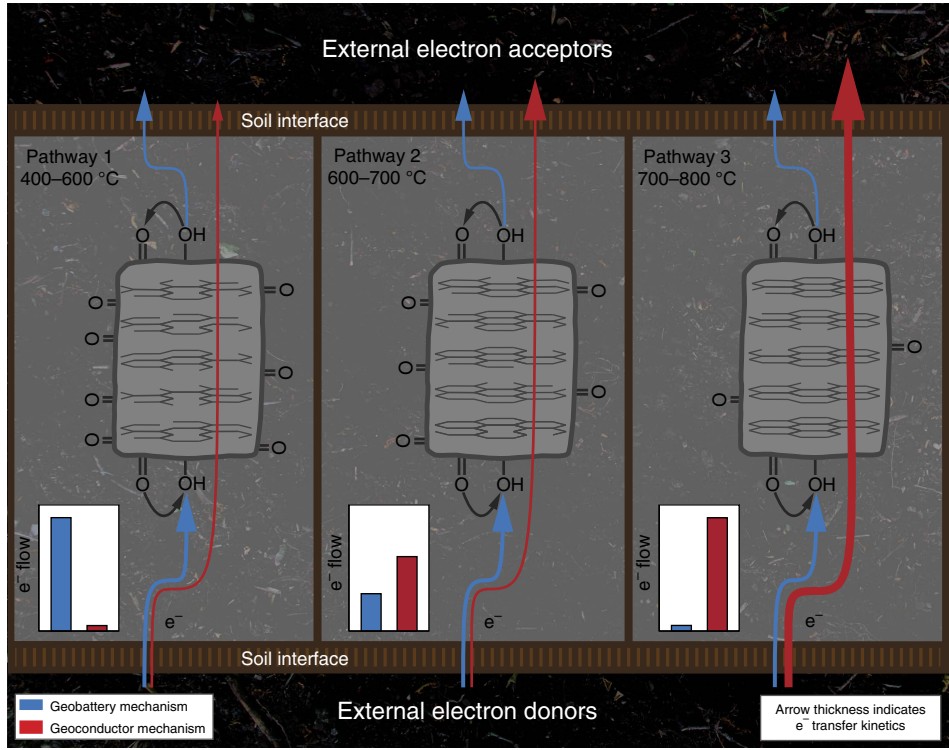

**Figure 5 | Schematic diagram of the pyrogenic carbon internal pathways for electron flows.** Blue arrows indicate the charging and discharging cycles of the geobattery mechanism; red arrows indicate the direct electron transfer through the geoconductor mechanism. Arrow thickness represents the magnitude of transfer kinetics. The dominating electron flow is illustrated in the inset chart for each pathway. The ordering of carbon structures determining the pathways 1, 2 and 3 are based on the Raman spectroscopy of pyrogenic carbon matrices.

The pyrolysis temperatures of this study cover the temperature range of forest and grassland fires. The average temperatures of a forest fire range from 300 to 800 °C, under extreme conditions such as the wood core burning, or peat burning, the temperature can exceed 1,000 °C. The general temperature profile during a forest fire follows the pattern that the temperature rapidly rises up to 800 °C then slowly decreases until it stops, and the fire dwell time depends on the area and extent of burning[46]. Longer periods of high temperature have been identified in many forest fires in Indonesia as a result of forest clearing and annual slash-and-burn interventions[47]. Temperatures up to 800 °C have also been measured in an oak ecosystem[48]. Temperatures of grassland fires are typically lower and range from about 200 to 550 °C although they follow similar patterns of temperature variation to forest fires[49]. The threshold temperature observed for the transition between electron transfer mechanisms (geobattery versus geoconductor) may vary from that reported in this study, as different starting biomass (chiefly as a result of its lignin and mineral contents) and pyrolysis conditions (duration, pressure, and oxygen availability) will generate different properties of pyrogenic carbon even under similar pyrolysis temperatures[50]. Rather, carbon structure and surface functionality (here determined by atomic H/C and O/C ratios and Raman, FTIR and EELS spectroscopy) provide the fundamental connection between organo-chemical properties and electron transfer properties of pyrogenic carbons.

Voltammograms of natural organic matter are usually feature-less (that is, with a lack of distinct current peaks) due to a combination of slow electron transfer kinetics and a distribution of reduction potentials of redox-active moieties[26,27,31]. The resolved charging and discharging peaks shown in our study (Fig. 1c,d) are probably due to the enrichment of quinone moieties in pyrogenic carbon and their relatively fast electron

transfer kinetics. These peaks also indicated that quinone moieties dominate the redox property of pyrogenic carbon both quantitatively and kinetically, although we could not completely rule out the contributions of other types of redox-active components that possess slow electron transfer kinetics and appear featureless in cyclic voltammograms. Similar to our study, Nurmi and Tratnyek[51] found that the polyphenolic moieties in organic matter have similar electrochemical behaviour as menadione and juglone by matching the formal potential, half-wave potential, and peak separations between polyphenolic moieties and various quinone model compounds as demonstrated by voltammetric scan in aprotic solvent. Spectroscopic results (Fig. 2) confirmed the presence of quinone groups and their decreased quantity with an increase in pyrolysis temperature and formation of graphitic carbon structures. Apart from quinone, other functional groups have also been observed to decrease in quantity in our study. Therefore, the featureless voltammogram of the high-temperature pyrogenic carbon is more likely due to the lack of redox-active components than unidentifiable potential distributions.

We argue for the benzo/hydroquinone couple as the most likely model compound because of the correspondence of the peak potentials, identical electron transfer kinetics and spectro-scopic consistency with the surface quinone groups of pyrogenic carbon. The benzo/hydroquinone couple was immobilized on a working electrode in a crystalized solid phase (Fig. 1c, the same condition as the pyrogenic carbon particles) rather than in a dissolved phase (Supplementary Fig. 7) as used when we performed the kinetic matching to surface quinone groups of pyrogenic carbon. Despite the fact that pyrogenic carbon can only form suspensions in solution, using the model benzo/hydroquinone couple in a dissolved phase enabled the direct comparison of kinetics between quinone groups and carbon

matrices of pyrogenic carbon (Fig. 1e). Although simple, this comparison provided a general sense of the competitive electron transfer pathway in pyrogenic carbon, that is, which surface property (quinone groups or structural carbon) will preferentially in kinetics transfer electrons donated from the same environmental interface. Better spatial alignment was likely occurring between the interface of the electrode and dissolved quinone species than that of the electrode and quinone groups on pyrogenic carbon particles, which potentially overestimated the electron transfer rate of quinone groups on pyrogenic carbon and would only make the argument stronger that surface quinone groups are less important at high temperatures than electron transfer through pyrogenic carbon matrices. Different working electrodes (or different pairs of redox species and working electrode) and different surface treatments will result in different electron transfer rates. In the case of quinones, polished glassy carbon electrodes possess similar electron transfer rates to platinum electrodes and are faster than diamond, highly ordered pyrolytic graphite, or mercury electrodes[32,52–54]. Surface treatments by decreasing the surface oxides coverage of glassy carbon has been shown to slightly improve the electron transfer kinetics of methylcatechol but no effect has been observed on benzo/hydroquinone couples[55]. Casting of carbon nanotubes on a glassy carbon surface did not show significant improvements of electron transfer kinetics of quinone compounds although the specific surface area increased[56]. Therefore, we used polished (aluminium oxide polishing powder, CHI polishing kit) glassy carbon as the working electrode in this study (Supplementary Fig. 7) to provide an estimation of fast/upper limit electron transfer kinetics of quinone compounds, and compared this kinetic to the direct electron transfer of pyrogenic carbon matrices (Fig. 1e).

It is expected that increasing amounts of pyrogenic carbon will accumulate in the environment due to predicted increases in fire frequency as a result of climate change[57] and intentionally added biochar for improving soil fertility and carbon sequestration[58]. Comprehensive understanding of the electron transfer mechanisms in pyrogenic carbon will help to better understand its fundamental roles in redox-driven biogeochemical cycles. While the mediated electron transfer model as a result of functional groups has been well established over the past 20 years[59], direct electron transfer and its demonstrated fast kinetics suggest that the electron flux through a conductive network of ubiquitous pyrogenic carbon needs to be considered when trying to close the charge balances for transformations of redox species in natural systems. Natural pyrogenic carbons can have sizes of millimetres to centimetres, and in contrast to the localized functional groups, the electron transfer by carbon matrices may lead to a relatively long-distance transport that provides a spatially more extensive accessibility to alternative electron acceptors such as minerals for anoxic microbial respiration, and could therefore have hitherto unappreciated effects on suppression of greenhouse gas emissions. Before microbial colonization, the access to or sensing of pyrogenic carbon by microbes is probably the rate-limiting step for electron transfer, and even a rapid electron transfer by carbon matrices will not significantly improve the overall extracellular electron transfer rate. However, knowing the electron transfer kinetics and pathways of pyrogenic carbon will become more important after microbes colonize pyrogenic carbon. It has been demonstrated that pyrogenic carbon is a favourable habitat for many microorganisms[60] and changes of the microbial community and activity are more pronounced by addition of high- than low-temperature pyrogenic carbon[61]. Our results may lead to discoveries that explain these microbial responses, such as the formation of microbial fuel cell structures in micro-pores of

pyrogenic carbon that allows rapid access to adsorbed electron acceptors for colonized microbes[62]. Future efforts should focus on the importance of the demonstrated electron transfer for microbial metabolism and its long-term effects on greenhouse gas emissions and metal biogeochemistry.

## Methods

**Chemicals and materials.** All chemicals were used as received unless otherwise noted. Potassium hexacyanoferrate(III) ([Fe(CN)$_6$]$^{3-}$, min. 99%), hexa-ammi-neruthenium(III) chloride ([Ru(NH$_3$)$_6$]$^{3+}$, 98%), (dimethylaminomethyl)ferrocene (FcDMAM, 96%), iron(III) chloride (97%), hydroquinone (HQ, min. 99%), anthraquinone-2-sulfonic acid (AQS, 97%), Nafion perfluorinated resin solution (5 wt. %), graphite rod (6 mm diameter, 99.999%), graphite powder (<45 μm, min. 99.99%) and Manganese(IV) oxide (10 μm, min. 90%) were obtained from Sigma-Aldrich (St Louis, MO). 1,4-benzoquinone (BQ, 97%) and 1,4-naphthoquinone (NQ, 97%) were purchased from TCI America and Alfa Aesar, respectively. Strem Chemicals (Newbury Point, MA) was the supplier of iron(II,III) oxide (magnetite, min. 95%) and iron(III) oxide (hematite, 99.8%-Fe). The measured Brunauer–Emmett–Teller areas (N$_2$) of manganese(IV) oxide, hematite and magnetite are $108 \pm 0.5$, $10 \pm 0.2$ and $7.7 \pm 0.1$ m$^2$ g$^{-1}$, respectively. Glassy carbon electrodes (3 mm diameter) and an electrode polishing kit were purchased from CH Instruments (Austin, TX). Millipore water (18 MΩ cm) was used to clean all glassware and prepare all aqueous solution.

**Preparation of pyrogenic carbon samples.** Well-characterized pyrogenic carbon samples were produced under controlled conditions in the laboratory by anoxic pyrolysis of woody biomass (black walnut). Pyrolysis temperatures were 400–800 °C. Dwell time was 30 min for all pyrolysis temperatures except for 650 °C (1 h). Pyrogenic carbon produced at 650 °C is in the transition region of slow to fast electron transfers of the studied carbon matrices (Fig. 1e). The redox peaks appeared very noisy and unstable when 30 min dwell time was applied. Therefore, we applied a slightly longer dwell time to improve the stability of the direct electron transfer by pyrogenic carbon produced at 650 °C. The influence of a longer dwell time on the calculation of rate constants is minor and only account for <1% of overpotential decrease and <2% of ln(current) increase, which is still in the range of the s.d. of replicate tests. The pyrolysis facility is equipped with a programmable Barnstead/Thermolyne muffle furnace using a cyclic argon protection system.

**Fabrication of pyrogenic carbon rod electrodes.** After pyrolysis, the pyrogenic carbon particles were fabricated into rods (5 mm diameter, 10 mm length) with the face following the wood grain as the electrochemical working surface. The working surface was carefully polished using a sand paper provided by the CH Instrument electrode polishing kit (Austin, TX). The resulting pyrogenic carbon rods were soaked in water and washed by several cycles of vacuum to remove the debris from mechanic fabrication and polishing, followed by drying at 105 °C overnight. The pyrogenic carbon rods were inserted into a polytetrafluoroethylene tube connected to the sharp end of thin copper rods reinforced with a spring. Vacuum deoxygenation (1 h) was performed to all pyrogenic carbon rod electrodes before electrochemical measurements.

**Immobilization of pyrogenic carbon on graphite electrodes.** Nafion was used as the binding reagent in immobilizing the pyrogenic carbon powders on graphite rod electrodes for electrochemical tests of surface quinone groups. Eight milligrams of solids and 40 μl of Nafion were dispersed into 1 ml of a mixture of ethanol and water (v:v 1:4) using a vortex. Afterwards, 15 μl of the resulting dispersion was dropped onto the surface of the working electrodes and dried at room temperature. All pyrogenic carbon powders were obtained by ball milling the pyrogenic carbon particles for 3 min, and adjusted to pH 7 with sodium hydroxide or hydrogen chloride in water media (containing 10% ethanol to reduce the surface tension of pyrogenic carbon) before immobilization on the working electrodes. All graphite electrodes with immobilized pyrogenic carbon were vacuum deoxygenated for 1 h before electrochemical measurements.

The reduction and oxidation peaks shown in Fig. 1c and Supplementary Fig. 5a clearly confirmed the feasibility of cyclic voltammetry for studying the charging and discharging behaviours of surface quinone groups in pyrogenic carbon immobilized on a graphite working electrode. We used wood biomass as feed stock for preparation of pyrogenic carbon due to its low ash, mineral and leachable organic compound content after pyrolysis[50], which minimized the contribution of unstructured redox-active species to the charging and discharging cycles of surface quinone groups[13,14]. Increasing the amount of immobilized pyrogenic carbon resulted in an increased peak current and total charging and discharging capacity, but the sensitivity (expressed by capacity per gram of pyrogenic carbon) decreased possibly due to the screening of effective contact between surface quinone groups and the working electrode (inset in Supplementary Fig. 5a). The variation of sensitivity indicated that the same amount of pyrogenic carbon should be applied when comparing the charging and discharging capacities among different temperatures. The close redox behaviour between benzo- and hydroquinone (Supplementary Fig. 5b) suggested that the electron transfer of quinone groups

followed a two-electron two-proton mechanism in this immobilized method with a pH-buffered supporting electrolyte. The cyclic voltammogram of immobilized naphthoquinone and anthraquinone-2-sulfonic acid for formal potential comparison are given in Supplementary Fig. 5c. Background voltammetric scans on graphite working electrode with Nafion and immobilized with graphite powder (particle size <45 μm) are shown in Supplementary Fig. 5d, which did not show featured peaks at around 0.05 V.

**Immobilization of minerals on pyrogenic carbon electrodes.** Minerals were immobilized on pyrogenic carbon rod electrodes using Nafion as binding reagent for electrochemical test of direct electron transfer from carbon matrices to minerals. Ten milligrams of minerals and 40 μl of Nafion were dispersed into 1 ml of a mixture of ethanol and water (v:v 1:4) using a vortex. Afterwards, 40 μl of the resulting dispersion was dropped onto the surface of the pyrogenic carbon electrode and dried at room temperature. Immobilization of iron(III) chloride was achieved by recrystallization of 40 μl of 3 mM FeCl$_3$ solution (containing 1.6 μl Nafion) on surfaces of the pyrogenic carbon rod electrodes. Fresh mineral samples were used at each scan rate in linear sweep voltammograms of mineral reduction. All pyrogenic carbon electrodes with immobilized minerals were vacuum deoxygenated for 1 h before electrochemical measurements.

**Electrochemical measurements.** All electrochemical measurements (cyclic voltammetry, linear sweep voltammetry and chronoamperometry) were performed in a three-electrode configured cell using a Bio-Logic model VSP potentiostat (France) controlled by the EC-Lab platform at room temperature. The pyrogenic carbon rod, graphite rod and glassy carbon electrodes were carefully polished and used as working electrodes based on the different purposes described in the text. Ag/AgCl (saturated KCl) and graphite rods were used as reference and counter electrodes, respectively. KCl (0.1 M) with pH buffered at 7 by phosphate was used as the supporting electrolyte. Purified N$_2$ gas was used to purge the oxygen from solution before measurements, and the N$_2$ atmosphere was maintained over the solution during subsequent electrochemical measurements. Anodic (oxidation) and cathodic (reduction) currents were designated to a positive and a negative value, respectively, and the scan direction for all cyclic voltammograms was changed from oxidation to reduction. Detailed experimental designs for electrochemical tests can be found in Supplementary Table 3.

**Selection of redox couples for direct electron transfer.** The redox species used to describe the electron transfer kinetics of materials can be generally classified into three categories from low to high kinetics[63]. The first category includes mainly cyanide compounds such as hexacyanoruthenate ($[Ru(CN)_6]^{3-/4-}$), hexacyanoferrate ($[Fe(CN)_6]^{3-/4-}$) and octacyanomolybdate ($[Mo(CN)_8]^{3-/4-}$), the second category includes mainly ferrocene and its derivatives$^{+/0}$, and the third category includes compounds such as tris(2,2′-bipyridyl)ruthenium ($[Ru(bpy)_3]^{3+/2+}$), hexaammineruthenium ($[Ru(NH_3)_6]^{3+/2+}$) and methyl viologen$^{2+/1+}$ with an electron transfer rate constant higher than ferrocene compounds but in the same order of magnitude. During preliminary testing of three redox couples with relatively neutral formal potential from each category using the pyrogenic carbon electrode (Supplementary Fig. 1), we found that the ferrocene derivative (dimethylaminomethyl)ferrocene (FcDMAM) had the fastest electron transfer response and a formal potential falling into the middle of the pyrogenic carbon potential window. Therefore, we used FcDMAM to characterize the maximum electron transfer kinetics of pyrogenic carbon matrices. ($[Fe(CN)_6]^{3-}$ had the lowest electron transfer rate constant (0.004 cm s$^{-1}$ compared to 0.017 and 0.018 cm s$^{-1}$ for $[Ru(NH_3)_6]^{3+}$ and FcDMAM, respectively). The formal potential of $[Ru(NH_3)_6]^{3+}$ is too negative ( − 0.134 V versus Ag/AgCl compared to 0.388 V for FcDMAM and 0.256 V for $[Fe(CN)_6]^{3-}$) for the characterization of pyrogenic carbon matrices and the electron transfer rate constant is slightly lower than that estimated by FcDMAM.

**Calculation of $k^0$ of pyrogenic carbon matrices.** Supplementary Figure 2a–f shows the cyclic voltammograms of pyrogenic carbon matrices pyrolysed at different temperatures, based on which the direct electron transfer rate constants ($k^0$) were calculated. The electron transfer kinetics of pyrogenic carbon matrices were divided into two classes based on the potential difference between the cathode and anode peaks ($\Delta E_P$): quasi-reversible kinetics for pyrogenic carbon pyrolysed at 725–800 °C and irreversible kinetics for pyrogenic carbon pyrolysed at 650–700 °C, which leads to two methods of calculation.

For pyrogenic carbon with quasi-reversible kinetics, the Nicholson method was used to calculate the rate constants[64]. Following the Nicholson's working curve, the $\Delta E_P$ can be converted into a dimensionless kinetic parameter $\psi$ that is directly proportional to the reciprocal of the square root of scan rate $v^{-1/2}$. The rate constant can thus be calculated by equation (1):

$$k^0 = \frac{\text{Slope}}{\sqrt{RT/(\pi nFD)}} \qquad (1)$$

where $k^0$ is given in cm s$^{-1}$, the slope is obtained from a linear fit of the $\psi \sim v^{-1/2}$ relationship (Supplementary Fig. 2g–i), $n$ is the number of electrons transferred

($n = 1$), $F$ is the Faraday constant and $D$ is the diffusion coefficient of FcDMAM in aqueous media ($D = 5.1 \times 10^{-6}$ cm$^2$ s$^{-1}$).

For pyrogenic carbon with irreversible kinetics, the rate constant was calculated using equation (2) based on the relationship of the natural logarithm of the absolute value of the cathode peak current ($\ln|i_{PC}|$) and overpotential (difference between cathode peak potential ($E_{PC}$) and formal potential ($E^{0'}$)) at different potential scan rates in the Tafel plots[64].

$$k^0 = \frac{e^{\text{Intercept}}}{0.227nFAC^*} \qquad (2)$$

Here, the intercept is obtained from a linear fit of the relationship $\ln|i_{PC}| \sim (E_{PC} - E^{0'})$ (Supplementary Fig. 2j–l), $A$ is the surface area of electrode (0.196 cm$^2$) and $C^*$ is the bulk concentration of FcDMAM in solution (3 mM).

Slower scan rates were applied with low-temperature pyrogenic carbon electrodes than with high-temperature pyrogenic carbon electrodes to properly capture the redox peaks for the estimation of kinetics. As can be seen from Fig. 1a,b and Supplementary Fig. 2a–f, the redox peaks increasingly separated towards lower pyrolysis temperatures, which implied decreased electron transfer kinetics by pyrogenic carbon matrices. This is caused by the increased inner resistance (indicated by the linear current–potential relationship between the redox peaks) and the less ordered pyrogenic carbon structures. The electron transfer kinetics of low-temperature pyrogenic carbon matrices cannot handle fast scan rates and the redox peak will otherwise be significantly diminished. For the kinetic assessments of this study, a slow scan rate therefore helped to better identify the redox peaks of low-temperature pyrogenic carbons, and it will not influence the quantification of said kinetics due to the linear relationship of ln(current) and potential on the Tafel plot for irreversible electron transfer behaviours (Supplementary Fig. 2j–l). Theoretically a fast scan rate will also fall into this linear region with the same slope and intercept, but the expression of redox peaks will not be as clear as when using a slow scan rate, or may even vanish due to the high overpotential while attempting to maintain the electron transfer at a fast scan rate.

**Correction of inner resistance.** The inner resistance of pyrogenic carbon matrices ($R_{\text{Inner}}$, Supplementary Table 4) have been corrected in all rate constant calculations according to the method described in ref. 65, because the interference of inner resistance causes an underestimation of electron transfer kinetics by decreasing the peak current and increasing the overpotential that has the same apparent effect as kinetics in cyclic voltammograms[64].

The interference of inner resistance in pyrogenic carbon with quasi-reversible kinetics was minor, as most of the nonconductive impurities had been removed during the high-temperature pyrolysis processes, and only <10% of the kinetics loss was found without any correction of inner resistance. However, for the pyrogenic carbon matrices with irreversible kinetics, the interference of inner resistance could induce a significant change to the calculation of the rate constants. As given in equation (2), the rate constant was calculated based on peak current and peak potential which are determined by both the kinetics and inner resistance, and the contribution of kinetics and inner resistance is indistinguishable. Therefore, an accurate estimation of the rate constant cannot be achieved. Instead, we estimated the lower limit of the rate constant from the relationship shown in equation (3) (Supplementary Fig. 2j–l), which underestimated the rate constant by containing contribution of resistance in kinetics. We estimated the upper limit from the relationship shown in equation (4) (Supplementary Fig. 2j–l) where the peak current was replaced by a practical maximum peak current for pyrogenic carbon produced at 800 °C (inset in Supplementary Fig. 2j). This procedure overestimated the rate constant by assuming that the peak current will not decrease with slower kinetics. In both cases, the peak potential ($E_{PC}$) was only corrected by the measured current using equation (5).

$$\ln\left|i_{\text{PC,Measured}}\right| \sim E_{\text{PC,Corrected}} - E_{\text{obs}}^{0'} \qquad (3)$$

$$\ln\left|i_{\text{PC,PyC800}}\right| \sim E_{\text{PC,Corrected}} - E_{\text{obs}}^{0'} \qquad (4)$$

$$E_{\text{PC,Corrected}} = E_{\text{PC,Measured}} \pm i_{\text{PC,Measured}} R_{\text{Inner}} \qquad (5)$$

The formal potential changed to more positive values with a decrease in pyrolysis temperature due to the increase of inner resistance (Fig. 1a), and a similar effect has been observed in cyclic voltammograms of $[Fe(CN)_6]^{3-}$ (Supplementary Fig. 13) with a trend to more negative values. To prevent overcorrection, we used the observed formal potentials (equation (6)) in $k^0$ calculations as shown in Supplementary Fig. 2j–l.

$$E_{\text{obs}}^{0'} = \frac{E_{PA} + E_{PC}}{2} \qquad (6)$$

The influence of the inner resistance on cyclic voltammograms of pyrogenic carbon with irreversible kinetics can be clearly seen in Supplementary Fig. 14. The partial removal of resistance by washing with either acid or organic solvents significantly increased the peak current and decreased the peak potential separation, but peak current or potential barely changed for those pyrogenic carbons with quasi-reversible kinetics. After correction as described above, the rate constant remained identical suggesting that the inner resistance is not a factor that determines kinetics of pyrogenic carbon. Pyrogenic carbon paste electrodes used in Supplementary

Fig. 14 were made by mildly grinding pyrogenic carbon powders after different treatments with inert paraffin oil (80:20 wt.) and pelleting 0.2 g of each paste into a hard plastic tube (5 mm diameter) using standardized pressure (191 kg cm$^{-2}$). Back contact was ensured by using a spring-pressed copper rod.

**Calculation of $k^{0'}$ of surface quinone groups.** From the cyclic voltammograms with immobilized pyrogenic carbon as redox-active species (Supplementary Fig. 3a–f), the charging and discharging rate constants ($k^{0'}$) of surface quinone groups and benzoquinone were calculated using the Laviron method[66]. With a peak potential differences of less than 0.2 V and transfer coefficient ($\alpha$) of 0.5 (approximated from the symmetric extension of reduction and oxidation peak potential as a function of logarithm of scan rate), the peak potential difference can be converted into a dimensionless kinetic parameter $m^{-1}$, which is directly proportional to the scan rate $v$, using equation (7):

$$k^{0'} = \frac{F}{RT\text{Slope}} \quad (7)$$

where $k^{0'}$ is in s$^{-1}$ and the slope is obtained from a linear fit of the $m^{-1} \sim v$ relationship (Supplementary Fig. 3g).

$$C = \frac{\int \frac{i_{PA}}{F} dt}{10^6 m} \quad (8)$$

The charging and discharging capacity ($C$, μmol e$^-$ (g pyrogenic carbon)$^{-1}$) of surface quinone groups was calculated by integrating the cathodic peak current ($i_{PA}$, baseline corrected,) and the reaction time ($t$) based on the potential scan rate, that is, the Faradaic area, divided by the mass of pyrogenic carbon ($m$), using equation (8). The cathodic peak current was used because of its clearer baseline than the anodic peak current; an identical Faradaic area of anodic and cathodic peaks was assumed due to the reversible reaction.

**Electrochemical capacitance of pyrogenic carbon.** The electrochemical capacitance of pyrogenic carbon was determined from the non-Faradaic current response to the potential scan in a supporting electrolyte containing 0.1 M KCl and using pyrogenic carbon as the working electrode (Supplementary Fig. 9a). The capacitance ($C_{capacitance}$) was calculated by equation (9):

$$C_{capacitance} = \frac{i_{capacitance}}{Av} \quad (9)$$

where $C_{capacitance}$ is in mF cm$^{-2}$, $i_{capacitance}$ (A) is the capacitive current determined by re-plotting the cyclic voltammogram to the current–time form as shown in Supplementary Fig. 9b, $A$ is the geometric surface area of the pyrogenic carbon working electrode (0.196 cm$^2$), and $v$ is the scan rate (V s$^{-1}$). Due to the inner resistance and linear current–potential relationship shown in the cyclic voltammogram, the capacitive current at steady state could not be identified on 600 °C pyrogenic carbon, and its capacitance was approximately estimated using the current separation at mid-point potential (inset chart in Supplementary Fig. 9b).

All direct electron transfer, charging and discharging rate constants, and capacitance analysis mentioned in this study are calculated by the methods described by equations (1–9) except when otherwise noted.

**FTIR and EELS spectroscopy.** Attenuated total reflectance Fourier transform infrared (ATR-FTIR) spectroscopy was performed using a Vertex 70 FTIR spectrometer (Bruker Corp., Billerica, MA) equipped with a deuterated L-alanine-doped triglycine sulfate (DLaTGS) detector. A Pike GladiATR accessory with a single-bounce diamond internal reflection element (IRE) and flow cell (Pike Technologies Madison, WI) was used for sampling. Complete temperature series of pyrogenic carbon samples were scanned 128 times in the mid-infrared region from 4,000 to 600 cm$^{-1}$ with a resolution of 2 cm$^{-1}$.

Pyrogenic carbon particles were dispersed onto a Copper TEM grid with a SiO support film. EELS was conducted on a FEI Tecnai F20 STEM/TEM (FEI Company, Hillsboro, OR) with a Gatan 865 HR-GIF spectrometer. The microscope was operated at 200 kV in scanning TEM mode with a $\sim$10 pA probe current. The resolution of the EELS spectra was 1.1 eV. Detailed peak assignments of FTIR and EELS spectra can be found in Supplementary Table 5.

**Raman spectral mapping.** Raman spectral mapping of pyrogenic carbon matrices was carried out using a Renishaw InVia confocal Raman microscope with a 532 nm wavelength laser source. Samples of pyrogenic carbon at different pyrolysis temperatures were finely polished with diamond polishing paper for Raman spatial mapping (polishing procedure in Supplementary Table 6). The Raman spectra at each temperature is a sum of >600 single spectra taken over $\sim$25 × 25 μm areas to provide representative spectra with high signal-to-noise ratio. Raman spectra integrated over an area were more representative of the sample than one spectrum from a single point. Furthermore, integrating the intensities of the $D$ peak and $G$ peak over the area mapped using ImageJ CSI[67] allowed us to investigate whether there were any features in the areas mapped that were unique to either peak, which would indicate inhomogeneity in the structure (Supplementary Fig. 15). No unique features were identified, which suggested that the ordering of the carbon in the pyrogenic carbon samples was homogenous at μm resolution.

Inspection of the polished surfaces before and after acquisition of Raman spectra allowed us to verify that the carbon was not significantly damaged by the laser. Supplementary Figure 16 shows images of two regions of a 700 °C pyrogenic carbon sample before and after Raman spectrum acquisition. In Supplementary Fig. 16a, clear evidence of damage to the surface is visible after using high laser power to acquire Raman spectra. This suggests that the laser altered and damaged the structure of the pyrogenic carbon during acquisition, rendering the resulting Raman spectra unreliable. Supplementary Figure 16b shows more moderate damage at a lower laser power. Supplementary Figure 16c shows that further reducing laser power allowed us to acquire maps without damaging the surface and altering the structure of the sample, yielding more reliable Raman spectra. Verifying that the laser had not damaged the sample during acquisition would have been much more challenging if the Raman spectra had been acquired from an irregular powder surface. Surface roughness of pyrogenic carbon matrices for performing Raman mapping was <2 μm. Each spectrum was collected from 900 to 4,100 cm$^{-1}$ with a spectral resolution of 6 cm$^{-1}$, and an exposure time of 2 s per pixel. We selected an operating power for the laser that ensured no significant damage to the surface of the samples. For the 800 °C sample, the laser was operated at 20 mW, and for the other samples the laser was operated at 4 mW. All spectra were power law background subtracted using ImageJ CSI and normalized to the height of the $G$ band. The $D$ to $G$ band peak positions and intensity ratios were obtained by least squares fitting of a Breit–Wigner–Fano curve to the $G$ peak and a Lorentzian curve to the $D$ peak[37] using MATLAB.

**Effect of ageing on direct electron transfer.** The ageing process was simulated by oxidation of pyrogenic carbon using hydrogen peroxide ($H_2O_2$, 30%). The ageing simulated here generated molar O:C ratios[68] that are similar to those found after exposure in soil located at warm-humid climates[69] over 100 years. We used $H_2O_2$ for oxidation because except for $H_2O$ all other chemical species may potentially interfere with further electrochemical tests performed on pyrogenic carbon after the oxidation treatment. It can be seen from Supplementary Fig. 10 that the direct electron transfer kinetics of pyrogenic carbon matrices decreased after oxidation. In contrast, no decrease in kinetics was observed for graphite which implied that the carbon structures in the tested pyrogenic carbon are less resistant to natural ageing processes than that in the tested graphite. However, the decreasing rate and extent varied greatly for pyrogenic carbon pyrolysed at different temperatures. Generally, faster loss of kinetics was found in pyrogenic carbon pyrolysed at lower temperatures (34% loss of kinetic for pyrogenic carbon produced at 800 °C after 7 days of oxidation time and 100% loss for pyrogenic carbon produced at 550 °C after 1 day of oxidation time).

**Data availability.** The authors declare that the data supporting the findings of this study are available within the article, and its Supplementary Information files, and from the corresponding author upon reasonable request.

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

## Acknowledgements

This research was supported by NSF-BREAD (grant number IOS-0965336), USDA NIFA Carbon Cycles (2014-67003-22069), and NSF Graduate Research Fellowship (DGE-1144153). Any opinions, findings and conclusions or recommendations expressed in this material are those of the authors and do not necessarily reflect the views of the donors. We thank Kelly Hanley and Leilah Krounbi for assistance with sample analysis, Michael Schmidt for assistance in FTIR spectroscopy, Hector Abruña for discussion of electron transfer rate constant calculation, John Grazul for assistance with pyrogenic carbon polishing, and Christopher Umbach for assistance with Raman mapping. This work

made use of the Cornell Center for Materials Research Shared Facilities that are supported through the NSF MRSEC program (DMR-1120296). We are grateful for the constructive comments by two anonymous reviewers.

## Author contributions

T.S. and J.L. planned the research, T.S. conducted the electrochemical experiments, FTIR tests, and analysed the data, B.D.A.L., D.A.M. and T.S. performed the Raman and EELS spectroscopy and analysed the data. J.J.L.G., A.E. and L.T.A. assisted with measurements and interpretation, T.S., B.D.A.L. and J.L. wrote the paper, and all authors commented on the manuscript.

## Additional information

**Competing interests:** The authors declare no competing financial interests.

