## [Peer Review File · Nature Communications]

Reviewers' Comments:

Reviewer #1 (Remarks to the Author)

Review of manuscript:

"Rapid electron transfer by the carbon matrix in natural pyrogenic carbon"
by Sun et al., submitted to Nature Communications

The reviewer decided to provide three levels of review: A. general comments (integrative assessment), B. specific comments on some aspects of the work, and C. wording and clarity comments, which are least critical.

A. General comments

The presented manuscript is a very interesting and timely study on the electron transfer properties of pyrogenic carbon materials (later also referred to as "chars" by the reviewer for simplicity). In this work, the authors use a set of chars formed at different pyrolysis temperatures from one feedstock and supposedly under the same atmosphere in the pyrolysis reactor. The authors primarily use cyclic voltammetry and Raman spectroscopy to characterize the chars. Overall, this is a thought provoking manuscript that resulted in the reviewer spending more time on the review than the average. In principle, the reviewer fully agrees with the conceptual idea of the authors that chars may both have redox-active surface functional groups as well as conductive properties, and that the relative contribution of these functions change as the pyrolysis temperature changes. The group of the last author is known for such breakthrough conceptual studies in the soil sciences and biogeochemistry. This is another fine conceptual idea added by the group.

At the same time, the manuscript has been sent to Nature communications, which (the reviewer believes) requires a particularly critical evaluation (more than for other journals in the field). The reviewer attempted to provide such a critical review. The reviewer is familiar with basic electrochemical knowledge and may thus serve as a reference for the average reader targeted with this work. In light of this, it is possible that some of the questions related to the electrochemical techniques used reflect some ignorance on the side of the reviewer. Other questions may not.

The following list summarizes the more fundamental and general questions and comments that the reviewer has on the manuscript. For most of them, more details are provided in section B.

1. Is it possible to generalize the findings given that the authors worked with only one feedstock and apparently one pyrolysis condition (except variations in temperature)? The wording in the summary and elsewhere in the manuscript suggests that the authors believe 600°C is a critical threshold. The reviewer believes that this temperature is somewhat arbitrary as many factors will determine extent of condensation in the chars.
2. Related to 1 and to the implications of the work: what is the typical range of pyrolysis temperatures encountered in nature. It seems to the reviewer that these temperatures are more likely in the range of 300-500 than 600-800°C? In other words, how much of the "pyrogenic carbon" found in natural systems is of the type (high T) that is conducting?
3. The finding that conductance in pyrogenic carbon increases with increasing pyrolysis temperature is not new. In fact, it has previously been shown even in the environmental literature (e.g., on the catalytic reduction of organic pollutants on the surfaces of pyrogenic carbon; see one citation below). While the authors use a different approach to measuring conductance as compared to previous studies, the finding itself is not strictly novel. This is also true for a decrease in the amount of redox active moieties in chars with increasing pyrolysis temperatures, as recently shown in another electrochemical study (Klupfel et al, ES&T, 2015).
4. On the implication side, the authors invoke that high T char may have another "pathway for

electron flow" than low T char (geoconductor versus geobattery). Besides the fact that the idea that high T char is conducting is not new, the reviewer believes that high T char also has a capacitance (that is, it can store electrons in the condensed graphite-like structures). The "geobattery function" may thus be valid for both low and high T char. The reviewer also has some questions on how "capacitance" is assessed in the manuscript. The authors admit that the method employed is not really adequate to assess capacities (lines 70-72). Also, is the absence of current features in a voltammogram really a good criterion to exclude a capacity (small "geobattery" symbol in Figure 4 in high T chars).

5. Most of the work focuses on assessing the rates at which electrons are transferred to and from redox-active groups on the char surface (assessed by analyzing dissolved quinones, however) and through the chars. The reviewer, however, wonders whether the rates are really critical in the biogeochemical context. Is electron transfer in the matrix of higher T char the rate-determining step in biogeochemical reactions? The reviewer is skeptical. More likely, rates are limited by the transfer of redox active constituents to the surface of the chars and /or respiration by bacteria that can use the char as electron acceptor. So the provocative question may be raised: why are we interested in the rates of electron transfer in the first place if the processes occur over hours to days to months to years?

6. Overall, the reviewer recommends that the authors add a table to the Supplementary Information specifying which electrodes (both supporting working electrode and the materials added), which binders (NAFION), which dissolved redox active species were used, etc. At times it was very difficult (and time consuming) to search for this information in the manuscript and the SI.

7. The statement that conductive chars have a wide potential window over which they can participate in reactions is confusing to the reviewer. If a material is conducting, then it will be readily polarized to an applied potential. In other words, the concept of a "potential range" makes more sense for discrete redox active compounds with defined reduction potentials or for semiconductors.

Overall, the reviewer feels positive about the manuscript. The authors chose a new experimental approach that is more sophisticated and promising than ordinary chemical reduction and oxidation experiments. At the same time, the reviewer feels strongly that the above points (and the specific points below, some of which provide more details on the points above) need to be addressed before this manuscript can be considered for publication in Nature Communications. As said above, the critical points raised here reflect the high impact nature to which this manuscript was submitted.

B. Specific comments

Page 2

Line 17: see comment below on potential window

Line 20: The reviewer is surprised that the authors come to this very general conclusion (up to a pyrolysis temperature of 600°C) based on having analyzed a set of chars formed at different T but only from one feedstock, (supposedly) only under one atmosphere during the pyrolysis, and for one temperature. If stated like this, many readers will only remember 600°C ... without considering/being aware of other factors that will determine the char chemical properties. Needs to be stated more carefully.

Page 3

Line 42: It is not sufficiently clear based on the information here how measurements in Figure 1a were conducted. This is "diffusive cyclic voltammetry"? The reviewer is not familiar with this term (a quick check in google seems to support that this is not a very common term). The authors refer to the reduction and oxidation of a dissolved redox-active species in solution on the surface of an electrode prepared from the char materials?

Figure 1a: Why was the scan rate changed from 150 to 50 mV/sec when going from the 675°C char to the 650°C. Changing the scan rate will change the overall shape of the charging/discharging cyclic voltammogram as well as the shapes of potential current peaks on the

background.

Figure 1b: If the reviewer understands correctly, the authors have conducted CV experiments on electrodes prepared from chars pyrolyzed at different temperatures. The chars (pyrogenic carbon powders) were immobilized on a graphite electrode using Nafion, as described in section S5? (It would be very helpful if the authors provided a table in the SI which data was collected with which electrode setup). If this is correct, then the reviewer misses the following controls: (i) WE with pure Nafion, (ii) WE with Nafion and graphite powder (should not show current peaks at about 0.05 V).

Figure 1a and b: potentials should be reported vs SHE not Ag/AgCl

Figure 1b: Two additional questions come to mind: First, the voltammogram has a pronounced capacitive current contribution (the discrepancy in the i - E curves between the anodic and the cathodic sweeps). Is this capacitive contribution due to the WE carrying the pyrogenic carbon and Nafion or is this capacitive current (in part) due to the added chars. It seems that there is a systematic increase in the capacitive current with increasing $T_{\text{pyrolysis}}$. Unfortunately, the data in the insert is not easily interpretable as a higher scan rate was used for the 700 and 800 °C chars? (and hence the much larger capacitive currents)?

Second, it would have been very easy to test if the peak features were indeed due to quinones by varying the solution pH. Changes in pH would change the peak position of the quinone by -59 mV per pH. It seems that these measurements could have been conducted once the electrode was prepared?

Or is it impossible to do such measurement because the microenvironment in which the pyrogenic carbon is reduced is defined by the Nafion?

Page 4

Line 50: "surprisingly". This statement makes it sound as if has not been known that graphitic structures readily conduct electrons. But this has been established for quite some time. Even for charcoals electron conductance as a function of pyrolysis temperature has been described in the literature (e.g., Xu et al., ES&T, 2015, 49, 3419). This is a general point of "criticism": in the eyes of the reviewer, the paper would sell better if the authors had better acknowledged previous work. They certainly make it sound as if "fast electron transfer" (= conductance) through the char matrix is something novel. In the eyes of the reviewer, it isn't (at least conceptually it isn't; at the same time, the reviewer acknowledges that there are not that many papers on the topic).

Page 60: again, see Paper by Xu et al, ES&T, 2015, 49, 3419 and related literature on conductivity in pyrogenic and graphitic materials.

Page 5

Line 70-72: The explanation for smaller capacities does seem to make sense. How does contact affect the charging and discharging kinetics, however? The reviewer expects that overall rates of electron transfer (current) to the PC (pyrogenic carbon) increases as the contact areas increase?

Line 80: How do the authors rationalize that the current peak is consistent with benzoquinone/hydroquinone? Would one not expect a much broader current peak (or maybe even a featureless voltammogram) for heterogeneous materials such as pyrogenic carbons? This is similar to direct cyclic voltammetry of dissolved organic matter. These materials are result in featureless voltammograms, due to a combination of slow electron transfer kinetics and a distribution in reduction potentials of quinone moieties. In that sense: what if the high temperature chars have such a wide distribution in reduction potentials of surface quinones that there are no features in the voltammogram. It seems that the authors take the absence of such features as being indicative of the absence of surface quinones? For the reasons above, this may not be true. Only spectroscopic analysis of the surface functional groups would reveal if quinones are present or absent. It does not seem that the Raman Spectroscopy provided this data.

Line 81-82: The reviewer is confused here. The authors argue in the SI (section 7) that PC forms suspensions in solution and, hence, that the rates of electron transfer to and from surface

quinone/hydroquinone pairs are not readily accessible by "diffusive cyclic voltammetry". It seems that, based on the correspondence in peak potentials of the PC and benzoquinone / benzohydroquinone, the authors then use this dissolved species to assess the rate of electron transfer. Is this valid? It seems that electron transfer between a solid (electrode) and another species needs correct alignment between the two for electrons to be transferred. Such proper alignments are expected to be more favorable for a dissolved species than a solid (i.e., PC). If this is true, how can the authors then use the dissolved species to assess electron transfer to a solid PC? Similarly, does the rate of ET not also heavily depend on the working electrode used? That is, can the data provided in Figure S8 be really "only" interpreted as reflecting the rates of electron transfer to quinones? This seems to be an oversimplification. Had the authors used a different working electrode (BTW: Which one was used here? This information is hard to find) or treated the electrode differently, would they not have expected different electron transfer rates? Phrased differently: does the provided data really only describe a quinone property or rather a property of the quinone-working electrode pair?

Line 86: How do the authors rationalize a faster rate constant in the un-buffered than buffered system? Given that protons are exchanged, would one not expect that the reaction then is faster in a pH buffered system? What was the "pH" measured in the un-buffered solution?

Page 6

Line 94: While the motivation for using Raman spectroscopy to assess the content of "sixfold aromatic rings" (BTW: what are these? Units that contain at least six fused aromatic rings?), additional spectroscopic techniques that are more sensitive to surface oxygen content and speciation would have been helpful to provide additional information on the chemistry of O-containing surface functional groups (such as quinones).

Page 7

Line 116-117: The data presented in Figure 3 indeed suggest electron transfer from the pyrogenic carbon formed at 800°C to redox couples with very different reduction potentials (i.e., covering a range of approximately 1.5 V). The reviewer agrees. However, the authors state that the carbon matrix has a remarkably wide potential window (of 1.5 V). This statement, to the reviewer, is confusing, because: if one works with a conductive material, would one not expect the material to transfer electrons over a wide potential range? Phrased differently: would the authors not expect every conductive material (metal or graphite, ...) to reduce and oxidize redox couples with reduction potentials within the stability diagram of water? To the reviewer, the definition of a "potential range" only makes sense for a semiconductor or for materials that are not at all conducting but contain redox-active moieties or metals that are reduced/oxidized at very different potentials.

Line 125: "This reduction trend also ...". is this not stating the obvious? As the reduction potential of the oxidized reaction partner decreases and approaches the reduction potential of the reductant, the ΔE decreases and hence the ΔGET approaches zero? How is this a finding through work presented here?

Line 127: "Furthermore, the electrochemical behavior between surface quinone groups ...". Unclear to the reviewer what is meant here. Which PCs do the authors refer to and which "electrochemical behavior"?

Line 124: "Fe³⁺ ions": given the extremely poor solubility of ferric iron, the reviewer wonders if these were truly dissolved Fe³⁺+aq (unlikely) or whether they were complexed or precipitated (hydrolyzed) species?

Line 130: were the Fe oxide reductions also carried out in a phosphate solution? If so, it is likely that the rates of electron transfer were largely affected by inner sphere adsorption of phosphate onto the oxide surface. The authors should acknowledge this "artifact" if phosphate was present in solution (see line 269 which suggests that phosphate was present in all systems).

Line 131-132: The reported potentials (at pH7) should be compared to published reduction potentials of these iron oxides. While this may be tricky for magnetite (due to its complex redox properties = structural FeII), it should be straightforward for hematite.

Line 131: it seems unlikely that the reduction proceeds to zero valent iron which should readily react with H⁺ under the formation of H₂. The authors should report reduction potentials of the Fe²⁺/Fe⁰ couple and compare these to the applied potentials. Also, should the voltammograms not show two waves in this case? One for electron transfer Fe³⁺/Fe²⁺ and a second wave at more negative potentials for Fe²⁺/Fe⁰?

Also, in Figure 3, why does Fe²⁺ show a reductive current? The authors believe that this is Fe²⁺ reduction to Fe⁰? How about proton reduction to H₂, catalyzed by Fe²⁺?

Line 137: It seems that the comparison to cable bacteria is a bit far fetched. Sure, cable bacteria are exciting. At the same time, the reviewer does not really see how this work relates to cable bacteria.

Page 8:

Line 139: There is no direct evidence for amorphous defects presented in this manuscript. The authors should either state which data they refer to or cite papers in support of this statement.

Line 145: "dramatically" (see above). Also, do the authors deduce the diminished capacity of the high-T materials from the disappearance of the current features in the voltammograms? If so, then there are two points to consider: First, it is very possible that voltammograms are featureless because ET to "surface redox active groups" is slow and if the groups cover a wide range in reduction potentials. The analogy (see above) is natural organic matter, which also has featureless voltammograms. Second, the reviewer is still a bit perplexed by the fact that the authors seem to have only integrated features on the background current trace. Does the background current trace not also contain information on the capacity of the material? Typically, one ignores this background "electrode charging and discharging" when analyzing dissolved redox active species. However, it seems that here the background current contains "capacitive" information of importance to the characterization of the solids? The reviewer would also like to draw the authors attention to a number of papers published in the material sciences that clearly show that electrons can be "stored" in carbonaceous solids (e.g., Konkanand and Kamat, J Phys Chem C, 2007, 111, 9012). Would the authors not expect that their high T materials can also act as a "capacitor"?

C. Wording and clarity comments

Page 2

General comment: The reviewer feels strongly that the "abstract" needs a revision to improve clarity

Line 6: Electron transfer reactions instead of Electron exchanges

Line 7: Can electron transfer reactions "participate" in transformations? Maybe better: affect, govern, are fundamental to ...

Line 12: why "however"? The reviewer seems to contradiction.

Line 12: were instead of are?

Line 12: not clear to the reviewer what is meant by "the valence changes between external ...". The moles of electrons transferred?

Line 14: not clear what is meant with "pathway" in this context. Pathway for what?

Line 16: ... than by the known ... (should a "by" be added?)

Line 17: not clear what is meant by the "internal carbon structure"

Line 18: "over" instead of "in" a wide redox potential range?

Line 19: unchanged with regards to what

Line 21: dramatically? This seem to be unusual terminology for a scientific paper

Page 3:

Line 26: instead of quinones maybe the authors should say quinone/hydroquinone pairs?

Line 29: the way written it sounds as if the production of GHG and metal cycles are direct part of the microbial metabolism (maybe true for GHG but not so obvious for metal cycles). Please revise to make clearer.

Line 31: the rate of electron transfer was not investigated. Add "of electron transfer" and change remains unclear

Line 36: "no information ... has been demonstrated". Awkward structure, please revise

Line 44: "by peak current and potential caused by ..." not clear what is meant. The height of the

peak currents and the potential at which current were highest?

Page 4:

Line 65: electron "terminals". Please rephrase, not clear what is meant by "terminals"

Page 5:

Line 74: "immobilized cyclic voltammetry": see comment above on diffusive cyclic voltammetry

Reviewer #2 (Remarks to the Author)

The paper reports on the conductive properties of pyrogenic carbon and how a transition from surface mediated quinone reduction to metallic-like conductivity occurs with increased temperature. While this may appear obvious (increase in graphite content with increasing temperature improves conductivity) the authors do a good job in characterising this transition, and there is an interesting link between the two different forms of electron transfer. On the whole this is an interesting paper and will be relevant to a wide range of scientists.

(1) It would have been useful to show the insets of figure 1a and 1b as full panels, as they are quite to compare to the main figures. The changes in scan rate make it harder to directly compare the different voltammograms, although maybe the changes weren't as significant?

(2) It would also be useful to mention the E_m for DMEAMF somewhere in the text or figure legend for figure 1.

(3) Line 51. The authors should probably use sigmoidal instead of 'S-shape'

(4) line 67. It is implied that quinone groups are responsible, can the authors provide evidence that it is quinone groups in this case (see also point 8)

(5) Line 106. Please define what is meant by 'the increase in ordering'

(6) Figure 2, panel B. I think the text and arrows in the figure are unnecessary - the figure shows a change in the relative intensities of the D and G peaks, the text is interpretation of the data and shouldn't be there.

(7) The authors measured the reactivity of pyrogenic carbon (pyrolysed at 800 °C) with a broad range of substrates (Figure 3). At this temperature the authors have indicated that the quinol content of the substrate would be minor. I'm not convinced that the assertion that electron transfer from pyrogenic carbon to mineral occurs via the quinol groups is fully justified. In this case would Mn and Fe³⁺ reduction occur more quickly with electrodes prepared at lower temperatures (with more quinol). How do the authors exclude the possibility that electron transfer is not occurring directly through pyrogenic carbon. One possibility is to reproduce the experiments using carbon electrodes prepared at lower temperatures, but the authors may be able to justify this by other means.

(8) In addition to the Raman spectra, has there been any attempt to quantify the amount of quinone species in each carbon sample? This would be very useful as a lot of the paper revolves around the role of quinone on the surface.

Responses to reviewers' comments

We are grateful to the reviewers for their favorable review of our work and the constructive comments and suggestions that will significantly improve the quality of this manuscript. We highly value the opportunity to adjust the manuscript and look forward to the reviewers' responses to our revision. In addition to a clean version of the thoroughly revised manuscript, any changes in response to reviewers' comments have also been highlighted in the change-tracked version of the revised manuscript main text and supplementary information. A clean version of all revised documents has also been provided to the reviewers for easier reading after revision. Below is a point by point response to the reviewers' comments.

Sincerely,
Tianran Sun (on behalf of authors)

Reviewers' comments:

Reviewer #1

The reviewer decided to provide three levels of review: A. general comments (integrative assessment), B. specific comments on some aspects of the work, and C. wording and clarity comments, which are least critical.

Author response: Accordingly, we responded the reviewer in three levels: A. general discussion, B. response to specific comments, and C. wording and clarification improvement.

A. General comments

The presented manuscript is a very interesting and timely study on the electron transfer properties of pyrogenic carbon materials (later also referred to as "chars" by the reviewer for simplicity). In this work, the authors use a set of chars formed at different pyrolysis temperatures from one feedstock and supposedly under the same atmosphere in the pyrolysis reactor. The authors primarily use cyclic voltammetry and Raman spectroscopy to characterize the chars. Overall, this is a thought provoking manuscript that resulted in the reviewer spending more time on the review than the average. In principle, the reviewer fully agrees with the conceptual idea of the authors that chars may both have redox-active surface functional groups as well as conductive properties, and that the relative contribution of these functions change as the pyrolysis temperature changes. The group of the last author is known for such breakthrough conceptual studies in the soil sciences and biogeochemistry. This is another fine conceptual idea added by the group.

At the same time, the manuscript has been sent to Nature communications, which (the reviewer believes) requires a particularly critical evaluation (more than for other journals in the field). The reviewer attempted to provide such a critical review. The reviewer is familiar with basic electrochemical knowledge and may thus serve as a reference for the average reader targeted with this work. In light of this, it is possible that some of the questions related to the electrochemical techniques used reflect some ignorance on the side of the reviewer. Other questions may not.

Author response: We greatly appreciate the reviewer's thorough and constructive review of our manuscript and the effort in providing many excellent comments and suggestions to improve this work. Further discussion will benefit both sides to better understand the electron transfer processes of pyrogenic carbon.

The following list summarizes the more fundamental and general questions and comments that the reviewer has on the manuscript. For most of them, more details are provided in section B.

1. Is it possible to generalize the findings given that the authors worked with only one feedstock and apparently one pyrolysis condition (except variations in temperature)? The wording in the summary and elsewhere in the manuscript suggests that the authors believe 600°C is a critical threshold. The reviewer believes that this temperature is somewhat arbitrary as many factors will determine extent of condensation in the chars.

Author response: We agree with the reviewer that many factors such as the type of feedstock, pyrolysis temperature, and pyrolysis duration may determine the physicochemical properties of the pyrogenic carbon. Our study utilized wood biomass and relatively short pyrolysis duration (30 min), and we cannot extrapolate the temperature-dependence of pyrolysed wood to other biomass, and had not intended to give that impression. We therefore revised this statement in the summary and elsewhere throughout the manuscript to be more specific in pointing out the wood biomass and pyrolysis duration. We do propose, however, that the connection between the carbon structure (here determined by molar H/C and O/C ratios and Raman spectroscopy) and electrochemical properties will hold, though, as the reviewer correctly pointed out, these carbon structures may form at different temperatures, depending on e.g. duration of heating and starting material (chiefly its lignin and mineral contents). This discussion has been added to the second paragraph of Discussion section in the main Manuscript.

The molar H/C and O/C ratios of pyrogenic carbon samples have been given in Supplementary Information Table S1 to replace the old weight ratios. The H/C and O/C ratios have also been added in Figure 1 and 3 of the main Manuscript to show a better relation to pyrolysis temperatures.

2. Related to 1 and to the implications of the work: what is the typical range of pyrolysis temperatures encountered in nature. It seems to the reviewer that these temperatures are more likely in the range of 300-500 than 600-800°C? In other words, how much of the "pyrogenic carbon" found in natural systems is of the type (high T) that is conducting?

Author response: The average temperature of a forest fire ranges from 300 to 800°C, under extreme conditions such as the wood core burning, or peat burning, the temperature can exceed 1000°C. The general temperature profile during a forest fire follows the pattern that the temperature rapidly rises up to 800°C then slowly decreases until it stops, and the fire dwell time depends on the area and extent of burning (Wotton et al., 2012). Longer periods of high temperature have been identified in many forest fires in Indonesia as a result of forest clearing and annual slash-and-burn interventions (Ketterings, 1999). Temperatures

up to 800°C have also been measured in an oak ecosystem (Alexis et al., 2007). Temperatures of grassland fires are relatively lower and range from about 200 to 550°C (Daubenmire, 1968; Bailey and Anderson, 1980) although they follow similar patterns of temperature variation compared to forest fires. This discussion has been added to the second paragraph of Discussion section in the main Manuscript. The legacy effects of different temperatures on pyrogenic carbon found in global soils are still poorly quantified; however, benzene polycarboxylic acid (BPCA) and other biomarker and spectroscopic techniques may be used to compare chars made under different controlled conditions and naturally occurring chars (Schneider et al., 2010), and e.g. indicate that bark from pine (a soft wood) exposed to wildfires produces char exposed to temperatures of 600-700°C (Schneider et al., 2013). There is bound to be high variability in fires, and bulk analyses of average temperatures may blur the temperature that the burning biomass is actually exposed to. Surely a topic ripe for exploration. Our results will incentivize such research as it makes clear how important it is to know what temperatures are responsible for producing the large amounts of char found in soils globally. One may also argue (though we did not include that in the manuscript) that high-temperature chars are proportionally more represented in pyrogenic carbon found in soils, as high temperature also produces chars that are more persistent in soils (see e.g. reference (Spokas, 2010) for a review and many other publications).

3. The finding that conductance in pyrogenic carbon increases with increasing pyrolysis temperature is not new. In fact, it has previously been shown even in the environmental literature (e.g., on the catalytic reduction of organic pollutants on the surfaces of pyrogenic carbon; see one citation below). While the authors use a different approach to measuring conductance as compared to previous studies, the finding itself is not strictly novel. This is also true for a decrease in the amount of redox active moieties in chars with increasing pyrolysis temperatures, as recently shown in another electrochemical study (Klupfel et al, ES&T, 2015).

Author response: We agree with the reviewer that the increased “electrical conductivity” (please note that this is different from the property that we target in this paper) of pyrogenic carbon with greater temperatures is known and has been demonstrated in many aspects as the reviewer mentioned.

However, this study aims at revealing the surface electrochemistry of pyrogenic carbon matrices that is exposed to direct electron transfers, instead of the well-known bulk conductivity properties (Xu et al., 2013, 2015). Electrochemical studies provide a means of measuring electron transfer kinetics and interactions at the interface of electron donor/acceptor and pyrogenic carbon surfaces (this is different from the well-known electrical conductivity). We realize that the reviewers are well familiar with this distinction, and acknowledge that we had not made this important difference sufficiently clear in the manuscript, and we made a significant effort to make this clearer even to the non-specialist.

Because we targeted the electrochemistry or transfer kinetics at the interface, this study was designed to investigate how good or bad the arrangement of surface carbon atoms are at transferring electrons. In the case of pyrogenic carbon, indeed, the electrochemical property is correlated to the electrical conductivity, that is, high temperature pyrogenic

carbon has both high conductivity and a faster electron transfer rate and vice versa. In order to be able to distinguish between the two phenomena, we have corrected the contribution of conductivity/resistance during the quantification of the electron transfer rate of pyrogenic carbon pyrolysed at both low and high temperatures (please see Supplementary Information Section 5). Therefore, the electrochemical property demonstrated in this study can be described as the function of surface carbon structures. This is also why we put more emphasis on a surface scattering technique (Raman spectroscopy) to characterize the determination of surface carbon structures on electron transfer, instead of XRD or NAXAFS spectroscopy that are better able to characterize bulk structure.

Combined with the electron transfer by surface functional groups shown both in this study and by a more comprehensive previous study (Klüpfel et al., 2014) (we couldn't find the literature of Klüpfel et al. (2015), we assume the reviewer meant the biochar redox property study given by Klüpfel et al. published in 2014), the determination and modification of electron transfer pathways by the dynamic transition of surface properties (i.e., the transition from surface functional groups to surface carbon structures, and correspondingly from the geobattery mechanism to the geoconductor mechanism) has been demonstrated. We also use prefix "geo" to demonstrate the emphasis of the interfacial electron transfer process caused by both surface functional groups and carbon matrices. Based on this discussion, we revised the Introduction section of the main Manuscript by making much clearer the electrochemical focus of this study and how this differs from previously reported electron transfer studies that probed the electrical conductivity of pyrogenic carbon.

4. On the implication side, the authors invoke that high T char may have another "pathway for electron flow" than low T char (geoconductor versus geobattery). Besides the fact that the idea that high T char is conducting is not new, the reviewer believes that high T char also has a capacitance (that is, it can store electrons in the condensed graphite-like structures). The "geobattery function" may thus be valid for both low and high T char. The reviewer also has some questions on how "capitance" is assessed in the manuscript. The authors admit that the method employed is not really adequate to assess capacities (lines 70-72). Also, is the absence of current features in a voltammogram really a good criterion to exclude a capacity (small "geobattery" symbol in Figure 4 in high T chars)?

Author response: We agree with the reviewer that "electrical conductivity" through pyrogenic carbon is not new, however, as we argue at length above, electrochemical interface kinetics in the pyrogenic carbon matrices are new. We do not repeat the arguments here, as they are mentioned in detail in response to Comment 3 above.

We are glad that the reviewer highlighted the capacitance mechanism caused by the graphite-like sheet structures in pyrogenic carbon. We did test this mechanism and found an increased capacitance with an increase in pyrolysis temperature. We decided not to include this mechanism because this study was more focused on the kinetic comparison of electron transfer between functional groups and carbon matrices. We agree with the reviewer that this mechanism cannot be excluded and therefore added this information to

the revised version (please see Metal electron acceptor section in the Results of main Manuscript and Supplementary Information Fig. S13). However, we propose to use a separate term (Geocapacitor) to discuss the capacitance mechanism instead of adding it into the geobattery mechanism. Following the terminology of electrochemistry, we propose to distinguish the mechanisms as follows: a geobattery stores electrons by chemical reactions of functional groups, a geocapacitor stores electrons by graphite-like carbon sheet structures without reactions, and a geoconductor directly transfers electrons without storage and release (or charge and discharge). The kinetics of the electron storage and release process of the geocapacitor mechanism is expected to be closer to those of the geoconductor mechanism as they share carbon matrices as the electron transfer pathway instead of through the charging and discharging pathway of surface quinone groups in its geobattery mechanism. We estimated the capacitance by a traditional electrochemical method that is calculated based on the capacitive current and scan rate of the cyclic voltammogram in a pure KCl electrolyte and then normalized by the surface area (geometry area). The calculated electrochemical capacitance increases from 0.013 to 20 mF cm⁻² from pyrogenic carbon produced at 600 to 800°C (please see Supplementary Information Section 11).

Yes, the method we used in this study is not as appropriate as the mediated chronoamperometry method (Aeschbacher et al., 2010; Klüpfel et al., 2014) at assessing the capacities of geobattery, yet this method is effective in assessing the geobattery kinetics (and our study did not target the capacity of chars, since others already studied that). We did not exclude the geobattery capacity at high temperatures, but the geobattery capacity indeed significantly decreased with an increase of pyrolysis temperature, which is in agreement with a previous study (Klüpfel et al., 2014). The small geobattery contribution that we had included in Figure 4 (now Figure 5) was based on both of the kinetic and decreased capacity criteria.

5. Most of the work focuses on assessing the rates at which electrons are transferred to and from redox-active groups on the char surface (assessed by analyzing dissolved quinones, however) and through the chars. The reviewer, however, wonders whether the rates are really critical in the biogeochemical context. Is electron transfer in the matrix of higher T char the rate-determining step in biogeochemical reactions? The reviewer is skeptical. More likely, rates are limited by the transfer of redox active constituents to the surface of the chars and/or respiration by bacteria that can use the char as electron acceptor. So the provocative question may be raised: why are we interested in the rates of electron transfer in the first place if the processes occur over hours to days to months to years?

Author response: We strongly agree with the reviewer that before microbial colonization, the access to or sensing of pyrogenic carbon by microbes is probably the rate limiting step and not the electron transfer through pyrogenic carbon. Even the rapid electron transfer by carbon matrices will not significantly improve the overall extracellular electron transfer rate. However, knowing of the electron transfer kinetics and pathways of pyrogenic carbon will become more important after microbes colonized pyrogenic carbon. It has been demonstrated that pyrogenic carbon is a favorable habitat for many microorganisms (Vanek and Lehmann, 2015) and the modification of microbial community and activity is

more pronounced by addition of high temperature pyrogenic carbon (Khodadad et al., 2011). Possibly, our results may lead to discovery that explains these microbial responses, such as the formation of microbial fuel cell structures in micro-pores of pyrogenic carbon that allows rapid access to adsorbed electron acceptors (Joseph et al., 2013) as well as interspecies electron transfer (Chen et al., 2014) for colonized microbes. In engineering application, where the pyrogenic carbon can be pre-colonized by microbes, the characterization of the electron transfer rate by pyrogenic carbon will provide fundamental information in predicting their performance for example in anaerobic digestion (Chen et al., 2014; Mumme et al., 2014) and microbial fuel cells (Yuan et al., 2013; Huggins et al., 2014). Therefore, we propose further efforts should focus on the importance of the demonstrated electron transfer for microbial metabolism and its long-term effects on greenhouse gas emissions and metal biogeochemistry. This discussion of implication has been added in the last paragraph of the Discussion section in the main Manuscript.

6. Overall, the reviewer recommends that the authors add a table to the Supplementary Information specifying which electrodes (both supporting working electrode and the materials added), which binders (NAFION), which dissolved redox active species were used, etc. At times it was very difficult (and time consuming) to search for this information in the manuscript and the SI.

Author response: Thank you for the suggestion, we agree that this would facilitate following the experimental procedures. The requested table has been added (please see Supplementary Information Table S2).

7. The statement that conductive chars have a wide potential window over which they can participate in reactions is confusing to the reviewer. If a material is conducting, then it will be readily polarized to an applied potential. In other words, the concept of a “potential range” makes more sense for discrete redox active compounds with defined reduction potentials or for semiconductors.

Author response: We used the term “potential window” because we were trying to invoke the electrochemical concept that is used to characterize the non-faradaic behavior of various electrode-supporting electrolyte interfaces before redox reactions. But as the reviewer has pointed out, the potential range we determined for the pyrogenic carbon is by a series of redox reactions which actually resemble a faradaic behavior and are technically different from the determination of the electrochemical potential window sensu strictu. Therefore, the term “potential window” has been revised to “potential range” in the manuscript. Yes, a conductive material will be readily polarized to any potential that is applied. Yet the applied potential will shift (i.e., overpotential) to trigger redox reactions due to the catalytic nature of different electrode materials. What we were trying to demonstrate is that pyrogenic carbon possesses a well-performed surface catalytic property that allows it to induce many redox reactions over a wide potential range.

Overall, the reviewer feels positive about the manuscript. The authors chose a new experimental approach that is more sophisticated and promising than ordinary chemical reduction and oxidation experiments. At the same time, the reviewer feels strongly that the above points (and

the specific points below, some of which provide more details on the points above) need to be addressed before this manuscript can be considered for publication in Nature Communications. As said above, the critical points raised here reflect the high impact nature to which this manuscript was submitted.

Author response: Again, we are very grateful for the constructive comments and we hope that we were able to revise the manuscript appropriately.

B. Specific comments

Page 2

Line 17: see comment below on potential window

Author response: The response to this comment is given below.

Line 20: The reviewer is surprised that the authors come to this very general conclusion (up to a pyrolysis temperature of 600°C) based on having analyzed a set of chars formed at different T but only from one feedstock, (supposedly) only under one atmosphere during the pyrolysis, and for one temperature. If stated like this, many readers will only remember 600°C...without considering/being aware of other factors that will determine the char chemical properties. Needs to be stated more carefully.

Author response: We agree with the reviewer that a general conclusion focused on one temperature irrespective of feedstock, length of heating or atmosphere is not appropriate as many factors can influence the physicochemical property of pyrogenic carbon. We made two revisions: (1) we specified (here and throughout the manuscript) that the 600C refer to this study (with woody biomass and the specific thermal treatment); and (2) we made the connection between the organochemical properties (here determined by molar H/C and O/C ratios and Raman spectroscopy) and the electrochemical properties more visible (which was not as much done in the abstract as it should have). We believe we can construe a credible argument that while the 600C may likely be different for different feedstocks and production conditions (as correctly noted by the reviewer), it is these same conditions that determine the organochemical composition of the pyrogenic carbon that in turn determine the temperature at which electrochemical behavior changes. This should be a robust chain of argument and we are curious if the reviewer agrees.

Page 3

Line 42: It is not sufficiently clear based on the information here how measurements in Figure 1a were conducted. This is “diffusive cyclic voltammetry”? The reviewer is not familiar with this term (a quick check in google seems to support that this is not a very common term). The authors refer to the reduction and oxidation of a dissolved redox-active species in solution on the surface of an electrode prepared from the char materials?

Author response: Cyclic voltammetry can be generally classified into three categories based on the type of mass transport, which include (1) the hydrodynamic force maintained mass transport such as voltammetry on a rotating disk/ring electrode, (2) diffusion

maintained mass transport such as voltammetry on a stationary electrode, and (3) immobilized voltammetry (i.e., no mass transport process is involved). The second and third voltammetry are the techniques we used in this study, and used the terms “diffusive cyclic voltammetry” and “immobilized cyclic voltammetry” for distinction. For Figure 1a (now Figure 1 a and b), the reviewer is right that we tested the electron transfer rate of carbon matrices (i.e., the geoconductor mechanism) by cyclic voltammetry of a redox-active species dissolved in solution on electrodes that were constructed using pyrogenic carbon produced at low to high temperatures. We agree with the reviewer that the “diffusive cyclic voltammetry” is not a common term (it has also been called “diffusion-controlled cyclic voltammetry” (Andricacos and Ross, 1984)). Therefore, we addressed this issue by using the more common term “cyclic voltammetry” for both diffusion-controlled and immobilized techniques, but stated the difference including electrode material, redox species, and operation method in a new table listed in the Supplementary Information Table S2.

Figure 1a: Why was the scan rate changed from 150 to 50 mV/sec when going from the 675°C char to the 650°C.? Changing the scan rate will change the overall shape of the charging/discharging cyclic voltammogram as well as the shapes of potential current peaks on the background.

Author response: As it can be seen from Figure 1a and b, the redox peaks became more and more separated at lower pyrolysis temperatures, which implied a decreased electron transfer kinetics by pyrogenic carbon matrices. This is caused by the increased inner resistance (featured by the linear current-potential relationship between the redox peaks shown in the pyrogenic carbon below 650°C) and the less ordered pyrogenic carbon structures. Therefore, the scan rate (the parameter controlling the test of electron transfer kinetics) needs to be applied slower on low temperature pyrogenic carbon electrodes to properly capture the redox peaks for the estimation of kinetics. In other words, the electron transfer kinetics of low temperature pyrogenic carbon matrices cannot handle fast scan rates since by using fast scan rates the redox peak will be significantly diminished. We agree with the reviewer that a change of the scan rate will cause a shift of the voltammograms. However, for the kinetic assessment purpose of this study, a slow scan rate helped to better identify the redox peaks of low-temperature pyrogenic carbons, and it will not influence the quantification of the said kinetic calculation due to the linear relationship of $\ln(\text{current})$ and potential on the Tafel plot for irreversible electron transfer behaviors, i.e., a high scan rate will theoretically also fall into this linear region with the same slope and intercept, but the expression of redox peaks will not be as clear as when using a low scan rate, and will even vanish due to the high overpotential on attempting to maintain the electron transfer at a fast scan rate. This explanation of scan rate change has been added in the rate constant calculation part in the Supplementary Information Section 4.

Figure 1b: If the reviewer understands correctly, the authors have conducted CV experiments on electrodes prepared from chars pyrolyzed at different temperatures. The chars (pyrogenic carbon powders) were immobilized on a graphite electrode using Nafion, as described in section S5? (It would be very helpful if the authors provided a table in the SI which data was collected

with which electrode setup). If this is correct, then the reviewer misses the following controls: (i) WE with pure Nafion, (ii) WE with Nafion and graphite powder (should not show current peaks at about 0.05 V.

Author response: Yes, the reviewer correctly understood the immobilized cyclic voltammograms shown in Figure 1b (now Figure 1c and d). As suggested by the reviewer, a Table S2 has been added in the Supplementary Information. The requested control tests have also been provided in Figure S6d in Supplementary Information, which did not show current peaks at a potential of 0.05 V.

Figure 1a and b: potentials should be reported vs SHE not Ag/AgCl.

Author response: Figure 1 has been revised accordingly.

Figure 1b: Two additional questions come to mind: First, the voltammogram has a pronounced capacitive current contribution (the discrepancy in the i - E curves between the anodic and the cathodic sweeps). Is this capacitive contribution due to the WE carrying the pyrogenic carbon and Nafion or is this capacitive current (in part) due to the added chars. It seems that there is a systematic increase in the capacitive current with increasing T pyrolysis. Unfortunately, the data in the insert is not easily interpretable as a higher scan rate was used for the 700 and 800°C chars? (and hence the much larger capacitive currents)? Second, it would have been very easy to test if the peak features were indeed due to quinones by varying the solution pH. Changes in pH would change the peak position of the quinone by -59 mV per pH. It seems that these measurements could have been conducted once the electrode was prepared? Or is it impossible to do such measurement because the microenvironment in which the pyrogenic carbon is reduced is defined by the Nafion?

Author response: First, we agree with the reviewer that the increased capacitive current is possibly due to the increased capacitance of added pyrogenic carbon (the scan rate did not change in Figure 1b (now Figure 1c and d) and was kept at 100 mV/s for all pyrogenic carbon, but the scan rate in Figure 1a and b did indeed change from low- to high-temperature pyrogenic carbon since (as we responded above) Figure 1a and b show different measurements than those shown in Figure 1c and d). We found an increased capacitance with increased pyrolysis temperature, and this discussion has now been added to the manuscript as the “Geocapacitor” mechanism as we discussed in Comment A4.

Second, we performed the potential-pH dependent test (in response to the request by the reviewer) and found slopes of around -56 and -53 mV/pH for pyrogenic carbon produced at 450 and 500°C (please see Figure S7 in Supplementary Information), which is in agreement with the quinone compound tested in solution (-59 mV/pH). The shift is probably due to, as the reviewer pointed out, the lower control of the buffering electrolyte on the microenvironment protected by Nafion, especially at low pH.

*Line 50: “surprisingly”. This statement makes it sound as if has not been known that graphitic structures readily conduct electrons. But this has been established for quite some time. Even for charcoals electron conductance as a function of pyrolysis temperature has been described in the literature (e.g., Xu et al., ES&T, 2015, 49, 3419). This is a general point of “criticism”: in the eyes of the reviewer, the paper would sell better if the authors had better acknowledged previous work. They certainly make it sound as if “fast electron transfer” (= conductance) through the char matrix is something novel. In the eyes of the reviewer, it isn’t (at least conceptually it isn’t; at the same time, the reviewer acknowledges that there are not that many papers on the topic).
Page 60: again, see Paper by Xu et al, ES&T, 2015, 49, 3419 and related literature on conductivity in pyrogenic and graphitic materials.*

Author response: Please refer to the response to Comment A3, where we argue what exactly is new and how we made this clearer in the revised version. In brief, this study aims at revealing the surface electrochemistry of pyrogenic carbon matrices that is exposed to direct electron transfers across the interface (which is new), instead of the bulk electrical conductivity property (which is not new, as pointed out by the reviewer). In the studies given by Xu et al. (2013, 2015) and related literatures, the electron transfer was calculated from redox transformations of contaminants and was then correlated to the conductivity of pyrogenic carbon; but the electrochemistry of pyrogenic carbon that determines the interface interaction of molecules and pyrogenic carbon and interface electron transfer has not been investigated, i.e., in our opinion, electrical conductivity \neq electrochemistry. We revised the Introduction of the main Manuscript by emphasizing the interfacial electrochemical characterization of pyrogenic carbon and pointed out the difference from previous studies.

Page 5

Line 70-72: The explanation for smaller capacities does seem to make sense. How does contact affect the charging and discharging kinetics, however? The reviewer expects that overall rates of electron transfer (current) to the PC (pyrogenic carbon) increases as the contact areas increase?

Author response: We agree with the reviewer that the current will increase if the contact area increases, whereas the current density which is generally used to quantify the overall rate remains the same irrespective of the size of the contact area. We believe that a change of contact mechanisms may fundamentally affect the charging and discharging kinetics. In this study, the way we immobilized the pyrogenic carbon formed a contact for outer-sphere electron transfer (i.e., no specific bonding between the pyrogenic carbon and the working electrode). The inner-sphere electron transfer due to covalent bonding may induce a different behavior, but meanwhile the surface functional group properties of pyrogenic carbon will be modified and this is what we were trying to avoid.

Line 80: How do the authors rationalize that the current peak is consistent with benzoquinone/hydroquinone? Would one not expect a much broader current peak (or maybe even a featureless voltammogram) for heterogeneous materials such as pyrogenic carbons? This is similar to direct cyclic voltammetry of dissolved organic matter. These materials are result in featureless voltammograms, due to a combination of slow electron transfer kinetics and a distribution in reduction potentials of quinone moieties. In that sense: what if the high

temperature chars have such a wide distribution in reduction potentials of surface quinones that there are no features in the voltammogram. It seems that the authors take the absence of such features as being indicative of the absence of surface quinones? For the reasons above, this may not be true. Only spectroscopic analysis of the surface functional groups would reveal if quinones are present or absent. It does not seem that the Raman Spectroscopy provided this data.

Author response: We argued that the electrochemical behavior of surface quinone groups located on the pyrogenic carbon resemble benzo/hydroquinone functional groups by thermodynamically matching the apparent formal potential of the pyrogenic carbon with various model quinone compounds. Indeed, we strongly agree with reviewer that for such a complexity like dissolved organic matter, it is usually difficult to identify featured redox peaks on voltammograms due to a variety of redox active species and slow kinetics as the reviewer mentioned. However, it is still possible to resolve the current peak(s) during the voltammetric scan if the dissolved organic matter is specifically enriched with certain moieties and possesses fairly fast kinetics, for instance, the polyphenolic moieties (Nurmi and Tratnyek, 2002) and sulfur species (Orlović-Leko et al., 2016) in natural organic matter. Similar to our study, Nurmi and Tratnyek (2002) found that the polyphenolic moieties in organic matter have similar electrochemical behavior as menadione and juglone by matching the formal potential, half-wave potential, and peak separations between polyphenolic moieties and various model quinone compounds as evidenced in cyclic voltammograms. Therefore, the redox peaks shown in voltammograms in our study indicated that the quinone dominates (quantitatively and kinetically) the electrochemical characteristics of pyrogenic carbon, although we cannot completely rule out the contributions from trace metals or other inorganic elements. In response, we added “mainly” or “dominated by” to the appropriate places throughout the manuscript and added this discussion in Discussion section of the main Manuscript.

Yes, we used Raman spectroscopy only for the investigation of the aromatic carbon structure, and it cannot provide useful information of surface functional groups of pyrogenic carbon. Upon the reviewer’s request, we provided the Fourier transform infrared spectroscopic (FTIR) and electron energy loss spectroscopic (EELS) data of pyrogenic carbon to support our discussion on quinone electrochemical transitions (please see Figure 2 and Functional group composition section in the Result of the main Manuscript. Peak assignments of spectroscopic results have also been added to the Supplementary Information Table S4). Both spectroscopic results confirmed the presence of quinone groups and its decreased quantity with an increase in pyrolysis temperature. Besides quinone, other functional groups have also been observed to decrease in quantity. Therefore, the featureless voltammogram of the high-temperature pyrogenic carbon is more likely due to the lack of redox active components than the unidentifiable potential distributions.

Line 81-82: The reviewer is confused here. The authors argue in the SI (section 7) that PC forms suspensions in solution and, hence, that the rates of electron transfer to and from surface quinone/hydroquinone pairs are not readily accessible by “diffusive cyclic voltammetry”. It seems that, based on the correspondence in peak potentials of the PC and benzoquinone/hydroquinone, the authors then use this dissolved species to assess the rate of

electron transfer. Is this valid? It seems that electron transfer between a solid (electrode) and another species needs correct alignment between the two for electrons to be transferred. Such proper alignments are expected to be more favorable for a dissolved species than a solid (i.e., PC). If this is true, how can the authors than use the dissolved species to assess electron transfer to a solid PC? Similarly, does the rate of ET not also heavily depend on the working electrode used? That is, can the data provided in Figure S8 be really “only” interpreted as reflecting the rates of electron transfer to quinones? This seems to be an oversimplification. Had the authors used a different working electrode (BTW: Which one was used here? This information is hard to find) or treated the electrode differently, would they not have expected different electron transfer rates? Phrased differently: does the provided data really only describe a quinone property or rather a property of the quinone-working electrode pair?

Author response: We rationalized the benzo/hydroquinone couple as the model compound not only because of the correspondence of the peak potentials but more importantly by its identical electron transfer kinetics to the surface quinone groups of pyrogenic carbon. The benzo/hydroquinone couple was in a crystalized solid state immobilized on a working electrode (same condition as pyrogenic carbon particles) not in the dissolved phase (as the reviewer surmised; we now made the text clearer) when we performed the kinetic matching to surface quinone groups of pyrogenic carbon. We then used the dissolved quinone species to assess the rate constant of pyrogenic carbon quinone groups. However, we agree with the reviewer that a more perfect alignment was probably occurring between the interface of the electrode and dissolved species than that of the electrode and pyrogenic carbon particles, which potentially overestimated the electron transfer rate of quinone groups on pyrogenic carbon. This part has been revised and we acknowledge a possible overestimation of quinone kinetics by measuring the dissolved phase. If we overestimated quinone kinetics, this would only make our conclusions stronger, as we arrive at the argument that surface functional groups are less important at high temperatures than matrix electron transfer. This discussion has been added in the fourth paragraph of the Discussion section in the main Manuscript.

The reviewer is absolutely right that different working electrodes (or different redox species-working electrode pairs) and different surface treatments will result in different electron transfer rates. In the case of quinones, polished glassy carbon electrodes possess similar electron transfer rates to platinum electrodes and are faster than diamond, highly ordered pyrolytic graphite, or mercury electrodes (Hale and Parsons, 1963; Xu et al., 1998; Quan et al., 2007; Monge-Romero and Suárez-Herrera, 2013). Surface treatment by decreasing the surface oxides coverage of glassy carbon was shown to slightly improve the electron transfer kinetics of methylcatechol but no effect was observed for benzo/hydroquinone couples (DuVall and McCreery, 1999). Casting of carbon nanotubes on glassy carbon surface did not show significant improvements of electron transfer kinetics of quinone compounds although the specific surface area increased (Liu et al., 2010). Therefore, we used polished (aluminum oxide polishing powder, CHI polishing kit) glassy carbon as the working electrode in this study (i.e., Figure S9) since it can provide an estimate of the fast/upper limit electron transfer kinetics of quinone compounds, and we compared this kinetic to the direct electron transfer of pyrogenic carbon matrices. Although simple, this comparison provided a general kinetical sense of the competitive

electron transfer pathway in pyrogenic carbon, i.e., which surface property (quinone groups or structural carbon) will preferentially transfer electrons donated from the same environmental interface. This part has been revised and we added more explanation why we used glassy carbon as the working electrode to investigate the kinetics of quinone in the Supplementary Information Section 9. The information of glassy carbon used as working electrode has also been added in the main Manuscript.

Line 86: How do the authors rationalize a faster rate constant in the un-buffered than buffered system? Given that protons are exchanged, would one not expect that the reaction then is faster in a pH buffered system? What was the “pH” measured in the un-buffered solution?

Author response: The pH of the unbuffered solution was also adjusted to 7 by potassium hydroxide and hydrogen chloride. This information has now been added to the manuscript. We expected a faster electron transfer rate in the unbuffered solution based on the previous study that the protonation process is only partly involved in the redox reactions of quinones and demonstrated smaller peak potential separation in the pH neutral solution (Quan et al., 2007). The redox couple in the unbuffered solution is quinone and the mixture of quinone dianion and hydrogen bonded quinone dianion instead of quinone and hydroquinone in the buffered solution (please see Supplementary Information Section 9). Our cyclic voltammograms supported this mechanism that a faster rate constant was obtained in the unbuffered solution than in the buffered solution.

Page 6

Line 94: While the motivation for using Raman spectroscopy to assess the content of “sixfold aromatic rings” (BTW: what are these? Units that contain at least six fused aromatic rings?), additional spectroscopic techniques that are more sensitive to surface oxygen content and speciation would have been helpful to provide additional information on the chemistry of containing surface functional groups (such as quinones).

Author response: By “sixfold aromatic rings” we meant the aromatic ring structures in pyrogenic carbon. To avoid confusion we revised this sentence to “the *D* peak arises due to the disordered structure in sp^2 carbon atoms”.

We agree with the reviewer that additional spectroscopic data would be useful, and as discussed above, the FTIR and EELS spectra that reveal the transitions of functional groups have been added to Figure 2 in the main Manuscript, in response to the reviewer’s suggestion.

Page 7

Line 116-117: The data presented in Figure 3 indeed suggest electron transfer from the pyrogenic carbon formed at 800°C to redox couples with very different reduction potentials (i.e., covering a range of approximately 1.5 V). The reviewer agrees. However, the authors state that the carbon matrix has a remarkably wide potential window (of 1.5 V). This statement, to the reviewer, is confusing, because: if one works with a conductive material, would one not expect the material to transfer electrons over a wide potential range? Phrased differently: would the authors not expect every conductive material (metal or graphite ...) to reduce and oxidize redox

couples with reduction potentials within the stability diagram of water? To the reviewer, the definition of a “potential range” only makes sense for a semiconductor or for materials that are not at all conducting but contain redox-active moieties or metals that are reduced/oxidized at very different potentials.

Author response: We agree with the reviewer that the potential window is not the best choice of terminology in this case and can be misunderstood. And yes, we expected that conductive materials can transfer electrons in the stability diagram of water, although different kinetics will apply. As argued in the response to Comment A7, the purpose of testing the pyrogenic carbon matrices with different redox systems was to demonstrate that the pyrogenic carbon possesses a well-performed surface catalytic property that can cover many redox reactions in a wide potential range. The term “potential window” has been revised to “potential range” throughout the manuscript to avoid confusion.

Line 125: “This reduction trend also ...” is this not stating the obvious? As the reduction potential of the oxidized reaction partner decreases and approaches the reduction potential of the reductant, the ΔE decreases and hence the ΔG approaches zero? How is this a finding through work presented here?

Author response: We agree with the reviewer, and this sentence has been removed.

Line 127: “Furthermore, the electrochemical behavior between surface quinone groups ...” Unclear to the reviewer what is meant here. Which PCs do the authors refer to and which “electrochemical behavior”?

Author response: The electrochemical behavior refers to the different reduction potentials between pyrogenic carbon surface functional groups and various minerals. The production temperature of the pyrogenic carbon used here was 500°C. We compare the reduction potential among pyrogenic carbon and common minerals that are redox active and possibly contained in the pyrogenic carbon to confirm that these minerals did not contribute to the major redox feature shown in the voltammograms. We revised the sentence to improve clarity to “Further, the distinction of reduction potentials between surface quinone groups and metal minerals in solution strongly suggested that the metals contained in the pyrogenic carbon were not responsible for the electron transfer and that the detected charging and discharging property of pyrogenic carbon was therefore mainly a result of quinone moieties”.

Line 124: “Fe³⁺ ions”: given the extremely poor solubility of ferric iron, the reviewer wonders if these were truly dissolved Fe³⁺aq (unlikely) or whether they were complexed or precipitated (hydrolyzed) species?

Author response: The Fe³⁺ ions were obtained by simply dissolving FeCl₃ in deionized water, during which we did not observe any products that resemble precipitates. However, they were recrystallized after we immobilized them on the pyrogenic carbon electrode, so we agree with the reviewer that technically they were not free Fe³⁺ ions and we have revised it to FeCl₃ to be more specific.

Line 130: were the Fe oxide reductions also carried out in a phosphate solution? If so, it is likely that the rates of electron transfer were largely affected by inner sphere adsorption of phosphate onto the oxide surface. The authors should acknowledge this “artifact” if phosphate was present in solution (see line 269 which suggests that phosphate was present in all systems).

Author response: Yes, the Fe oxides were reduced in a phosphate buffered solution, and we thank the reviewer for providing the information that adsorbed phosphate can alter the reduction kinetics of Fe oxides. This part has been revised to reflect that.

Line 131-132: The reported potentials (at pH7) should be compared to published reduction potentials of these iron oxides. While this may be tricky for magnetite (due to its complex redox properties = structural FeII), it should be straightforward for hematite.

Author response: The comparison of reduction potentials obtained in this study with previously reported results have now been added to the text of manuscript as “Magnetite and hematite minerals showed a very similar reduction potential that ranged from -150 mV to -100 mV vs. SHE from a high to low scan rate, which was consistent with previously reported potentials (about -200 to -120 mV vs. SHE) obtained by thermodynamic calculations and microbial mineral reduction (Zachara et al., 2002; White et al., 2013; Gorski et al., 2016).”

Line 131: it seems unlikely that the reduction proceeds to zero valent iron which should readily react with H+ under the formation of H2. The authors should report reduction potentials of the Fe2+/Fe0 couple and compare these to the applied potentials. Also, should the voltammograms not show two waves in this case? One for electron transfer Fe3+/Fe2+ and a second wave at more negative potentials for Fe2+/Fe0? Also, in Figure 3, why does Fe2+ show a reductive current? The authors believe that this is Fe2+ reduction to Fe0? How about proton reduction to H2, catalyzed by Fe2+?

Author response: The reduction of FeCl₃ did show two current peaks in one voltammogram as the reviewer mentioned, one is Fe³⁺/Fe²⁺ and is followed by a second peak (Fe²⁺/Fe⁰) at more negative potential. We manually broke this voltammogram to insert the Fe oxides as their reduction potentials lay in between. The reduction of H⁺ ions cannot give such a high peak current at a pH of 7, we did not observe bubbles of a gaseous product, and more importantly this peak did not show up during the reduction of other species when we scanned the potential to the same negative value. Dissociation of water (i.e., the electrolysis of water) gives a continuous decrease in current until the potential scan is stopped or the potential is scanned back, but does not result in a current peak. Therefore, we suggest that the second peak was a result of Fe²⁺ reduction. We agree with the reviewer that Fe⁰ is very unstable and should be readily oxidized by H⁺, but the transport of H⁺ ions might be diminished by the Nafion film so that kinetically the net excess of Fe⁰ could occur in a situation of limited oxidation by H⁺ ions. This discussion has been added to the Metal electron acceptors section of Result section in the main Manuscript.

Line 137: It seems that the comparison to cable bacteria is a bit farfetched. Sure, cable bacteria are exciting. At the same time, the reviewer does not really see how this work relates to cable bacteria.

Author response: We agree with the reviewer and this sentence has been removed.

Page 8:

Line 139: There is no direct evidence for amorphous defects presented in this manuscript. The authors should either state which data they refer to or cite papers in support of this statement.

Author response: The reviewer is right that the Raman spectra did not provide direct evidence of “defects”. This sentence has been revised to “amorphous carbon structures” that was evidenced by the low D to G band ration shown in the Raman spectra.

Line 145: “dramatically” (see above). Also, do the authors deduce the diminished capacity of the high-T materials from the disappearance of the current features in the voltammograms? If so, then there are two points to consider: First, it is very possible that voltammograms are featureless because ET to “surface redox active groups” is slow and if the groups cover a wide range in reduction potentials. The analogy (see above) is natural organic matter, which also has featureless voltammograms. Second, the reviewer is still a bit perplexed by the fact that the authors seem to have only integrated features on the background current trace. Does the background current trace not also contain information on the capacity of the material? Typically, one ignores this background “electrode charging and discharging” when analyzing dissolved redox active species. However, it seems that here the background current contains “capacitive” information of importance to the characterization of the solids? The reviewer would also like to draw the authors’ attention to a number of papers published in the material sciences that clearly show that electrons can be “stored” in carbonaceous solids (e.g., Konkanand and Kamat, J Phys Chem C, 2007, 111, 9012). Would the authors not expect that their high T materials can also act as a “capacitor”?

Author response: Yes, we deduced the decreased geobattery capacity of high temperature pyrogenic carbon by the featureless current in cyclic voltammograms. As the reviewer suggested above, we did additional spectroscopic analysis (FTIR and EELS) of surface functional groups of pyrogenic carbon from low to high temperature. Both spectroscopic results confirmed the presence of quinone groups and its decreased quantity with an increase in pyrolysis temperature. Besides quinone groups, other functional groups have also been observed to decrease in quantity. Therefore, the featureless voltammogram of high temperature pyrogenic carbon is more likely due to the lack of redox active components than its unidentifiable nature.

We agree with the reviewer that the capacitance (caused by the graphite-like carbon structures and not the geobattery capacity caused by the redox cycles of surface functional groups) of pyrogenic carbon increased with the greater pyrolysis temperature. We also thank the reviewer for sharing the interesting study on the capacitive charging and discharging behavior of carbon nanotubes (Kongkanand and Kamat, 2007). We did expect the capacitor function of high temperature pyrogenic carbon, and in agreement with the

reviewer's suggestion, we have now added the voltammetric test of capacitance to the manuscript. As discussed above, we proposed to list the capacitance behavior of pyrogenic carbon as the "geocapacitor" mechanism instead of combining it within the geobattery mechanism.

C. Wording and clarity comments

Page 2

General comment: The reviewer feels strongly that the "abstract" needs a revision to improve clarity

Author response: Due to the editorial requirement, the abstract has been significantly rewritten and reduced from 217 to 150 words. We appreciate the reviewer's effort in improving the abstract and apologize the inconvenience. As the reviewer requested, the abstract has now been revised to be more specific.

Line 6: Electron transfer reactions instead of Electron exchanges

Author response: This expression has been revised accordingly and now been used as the first sentence of the introduction section.

Line 7: Can electron transfer reactions "participate" in transformations? Maybe better: affect, govern, are fundamental to ...

Author response: It has been revised to "govern" and is now used as the first sentence of the introduction section.

Line 12: why "however"? The reviewer seems to contradiction.

Author response: This sentence has been removed.

Line 12: were instead of are?

Author response: This sentence has been removed.

Line 12: not clear to the reviewer what is meant by "the valence changes between external ..." The moles of electrons transferred?

Author response: We intended to refer to the valence change, not the moles of electron transfer. Previous studies demonstrated the electron transfer by pyrogenic carbon using the valence change of exterior reactants (such as $\text{Fe}^{3+}/\text{Fe}^{2+}$ in iron minerals) as an indicator. In these cases the electron transfer pathways inside the pyrogenic carbon (i.e., whether the surface functional groups transferred the electrons or the carbon matrices) remained a black box and could not be identified. This sentence has been removed.

Line 14: not clear what is meant with "pathway" in this context. Pathway for what?

Author response: Pathway for electron transfer inside the pyrogenic carbon, i.e., whether the surface functional groups transferred the electrons or the carbon matrices. This sentence has been removed.

Line 16: ... than by the known ... (should a “by” be added?)

Author response: These words have been removed to shorten the Abstract.

Line 17: not clear what is meant by the “internal carbon structure”

Author response: It meant the carbon matrix relative to surface functional groups. This expression has been removed.

Line 18: “over” instead of “in” a wide redox potential range?

Author response: We revised it to “have a 1.5 V potential range”.

Line 19: unchanged with regards to what

Author response: With regard to charging and discharging kinetics. This sentence has been revised to reflect that.

Line 21: dramatically? This seem to be unusual terminology for a scientific paper

Author response: We agree, it has been removed.

Page 3:

Line 26: instead of quinones maybe the authors should say quinone/hydroquinone pairs?

Author response: We agree, it has been revised to quinone/hydroquinone pairs.

Line 29: the way written it sounds as if the production of GHG and metal cycles are direct part of the microbial metabolism (maybe true for GHG but not so obvious for metal cycles). Please revise to make clearer.

Author response: It has been revised and now reads “Such redox processes have been demonstrated to play important roles in suppression of greenhouse gas emissions, iron mineral reduction, and decontamination”

Line 31: the rate of electron transfer was not investigated. Add “of electron transfer” and change remains unclear

Author response: This sentence has been revised as suggested.

Line 36: “no information ... has been demonstrated”. Awkward structure, please revise

Author response: We agree, and this sentence has been removed.

Line 44: “by peak current and potential caused by ...” not clear what is meant. The height of the peak currents and the potential at which current were highest?

Author response: Yes, we referred to the height of the peak currents and the potentials at which this current as highest. This sentence has been revised as suggested.

Page 4:

Line 65: electron “terminals”. Please rephrase, not clear what is meant by “terminals”

Author response: Terminals meant acceptor and donator. This sentence has been revised to use these terms.

Page 5:

Line 74: “immobilized cyclic voltammetry”: see comment above on diffusive cyclic voltammetry

Author response: We agree with the reviewer that the “immobilized CV” is also not a common term. We have revised it by just using the term “cyclic voltammetry” and defined the operation condition in a new Table S2 in Supplementary Information.

Reviewer #2:

The paper reports on the conductive properties of pyrogenic carbon and how a transition from surface mediated quinone reduction to metallic-like conductivity occurs with increased temperature. While this may appear obvious (increase in graphite content with increasing temperature improves conductivity) the authors do a good job in characterizing this transition, and there is an interesting link between the two different forms of electron transfer. On the whole this is an interesting paper and will be relevant to a wide range of scientists.

Author response: We thank the reviewer for the supportive review of our manuscript and the effort in providing many constructive comments and suggestions to improve this work.

(1) It would have been useful to show the insets of figure 1a and 1b as full panels, as they are quite to compare to the main figures. The changes in scan rate make it harder to directly compare the different voltammograms, although maybe the changes weren't as significant?

Author response: The inset chart of Figure 1a and b is now shown as a full panel.

(2) It would also be useful to mentions the E_m for DMEAMF somewhere in the text or figure legend for figure 1.

Author response: The E_m (formal potential = 0.59 V vs. SHE) of FcDMAM (there is no chemicals abbreviated as DMEAMF, we assume the reviewer meant FcDMAM, i.e., (Dimethylaminomethyl)ferrocene) has been given in the caption of Figure 1.

(3) Line 51. The authors should probably use sigmoidal instead of 'S-shape'

Author response: This term has been revised to “sigmoidal” as suggested.

(4) Line 67. It is implied that quinone groups are responsible, can the authors provide evidence that it is quinone groups in this case (see also point 8)

Author response: To confirm the existence of quinone and reveal its transitions with increasing pyrolysis temperatures, additional Fourier transform infrared spectroscopic (FTIR) and electron energy loss spectroscopic (EELS) analyses of the pyrogenic carbon samples have been performed and added to the manuscript (please see Figure 2 and Functional Group Composition section in the Results section of the main Manuscript. Peak assignments of spectroscopic results have now been given in Supplementary Information Table S4). Both spectroscopic results demonstrated the featured peaks of quinone groups and its decreased quantity with an increase in pyrolysis temperature. In agreement with the electrochemical observation, the peak shoulders at 284.3 eV and characteristic peaks at 286.4 eV shown in the EELS spectra can be used to identify benzoquinone as the main surface quinone groups in low temperature pyrogenic carbon. Besides quinone, other functional groups have also been observed to decrease in quantity with an increase in temperature.

(5) Line 106. Please define what is meant by 'the increase in ordering'

Author response: We refer to the aromatic carbon ring structures who are more ordered with an increase in pyrolysis temperature. This sentence has been revised to read “Growth of ordered pyrogenic carbon structure”.

(6) Figure 2, panel B. I think the text and arrows in the figure are unnecessary - the figure shows a change in the relative intensities of the D and G peaks, the text is interpretation of the data and shouldn't be there.

Author response: The text and arrows in Figure 2b has been removed.

(7) The authors measured the reactivity of pyrogenic carbon (pyrolysed at 800 oC) with a broad range of substrates (Figure 3). At this temperature the authors have indicated that the quinol content of the substrate would be minor. I'm not convinced that the assertion that electron transfer from pyrogenic carbon to mineral occurs via the quinol groups is fully justified. In this case would MN and fe3+ reduction occur more quickly with electrodes prepared at lower temperatures (with more quinol). How do the authors exclude the possibility that electron transfer is not occurring directly through pyrogenic carbon. One possibility is to reproduce the experiments using carbon electrodes prepared at lower temperatures, but the authors may be able to justify this by other means.

Author response: We fully agree with the reviewer that the electron transfer is not occurring via the quinone groups, and we made that clearer by emphasizing the mineral reduction was occurring on pyrogenic carbon matrices. We used couples of redox active minerals with very different reduction potentials (from positive to negative potential) to

test the direct electron transfers via pyrogenic carbon matrices (i.e., Figure 3, which is now Figure 4), whereas surface quinones played a minor role, since the contribution of quinones in the high temperature pyrogenic carbon was minor. The purpose of this test was to demonstrate that the pyrogenic carbon matrices possess a well-performed surface catalytic property that can cover many redox reactions over a wide range of potentials. The reason we also tested the quinone groups here is to prove that the metals contained in the pyrogenic carbon were not responsible for the detected charging and discharging property of pyrogenic carbon due to the distinct reduction potentials between quinone groups and metal minerals shown in this figure.

(8) In addition to the raman spectra, has there been any attempt to quantify the amount of quinone species in each carbon sample? This would be very useful as a lot of the paper revolves around the role of quinone on the surface.

Author response: Please refer to the response to point 4 where we agreed and outlined that we now added significant new data that do exactly what the reviewer asks for. Thanks for the suggestion.

Reference cited

- Aeschbacher, M., Sander, M., Schwarzenbach, R.P., 2010. Novel Electrochemical Approach to Assess the Redox Properties of Humic Substances. *Environmental Science & Technology* 44, 87-93.
- Alexis, M.A., Rasse, D.P., Rumpel, C., Bardoux, G., Péchot, N., Schmalzer, P., Drake, B., Mariotti, A., 2007. Fire Impact on C and N Losses and Charcoal Production in a Scrub Oak Ecosystem. *Biogeochemistry* 82, 201-216.
- Andricacos, P.C., Ross, P.N., 1984. Diffusion-Controlled Multisweep Cyclic Voltammetry: III. Deposition of Silver on Stationary and Rotating-Disk Electrodes. *Journal of The Electrochemical Society* 131, 1531-1538.
- Bailey, A.W., Anderson, M.L., 1980. Fire Temperatures in Grass, Shrub and Aspen Forest Communities of Central Alberta. *Journal of Range Management* 33, 37-40.
- Chen, S., Rotaru, A.-E., Shrestha, P.M., Malvankar, N.S., Liu, F., Fan, W., Nevin, K.P., Lovley, D.R., 2014. Promoting Interspecies Electron Transfer with Biochar. *Scientific Reports* 4.
- Daubenmire, R., 1968. Ecology of Fire in Grasslands. in: Cragg, J.B. (Ed.). *Advances in Ecological Research*. Academic Press, pp. 209-266.
- DuVall, S.H., McCreery, R.L., 1999. Control of Catechol and Hydroquinone Electron-Transfer Kinetics on Native and Modified Glassy Carbon Electrodes. *Analytical Chemistry* 71, 4594-4602.
- Gorski, C.A., Edwards, R., Sander, M., Hofstetter, T.B., Stewart, S.M., 2016. Thermodynamic Characterization of Iron Oxide-Aqueous Fe²⁺ Redox Couples. *Environmental Science & Technology* 50, 8538-8547.
- Hale, J., Parsons, R., 1963. Reduction of P-Quinones at a Dropping Mercury Electrode. *Transactions of the Faraday Society* 59, 1429-1437.
- Huggins, T., Wang, H., Kearns, J., Jenkins, P., Ren, Z.J., 2014. Biochar as a Sustainable Electrode Material for Electricity Production in Microbial Fuel Cells. *Bioresource Technology* 157, 114-119.

Joseph, S., Graber, E.R., Chia, C., Munroe, P., Donne, S., Thomas, T., Nielsen, S., Marjo, C., Rutledge, H., Pan, G.X., Li, L., Taylor, P., Rawal, A., Hook, J., 2013. Shifting Paradigms: Development of High-Efficiency Biochar Fertilizers Based on Nano-Structures and Soluble Components. *Carbon Management* 4, 323-343.

Ketterings, Q.M., 1999. Fire as a Land Management Tool in Sepunggur, Sumatra, Indonesia, Can Farmers Do without It? Ohio State University.

Khodadad, C.L.M., Zimmerman, A.R., Green, S.J., Uthandi, S., Foster, J.S., 2011. Taxa-Specific Changes in Soil Microbial Community Composition Induced by Pyrogenic Carbon Amendments. *Soil Biology and Biochemistry* 43, 385-392.

Klüpfel, L., Keiluweit, M., Kleber, M., Sander, M., 2014. Redox Properties of Plant Biomass-Derived Black Carbon (Biochar). *Environmental Science & Technology* 48, 5601-5611.

Kongkanand, A., Kamat, P.V., 2007. Interactions of Single Wall Carbon Nanotubes with Methyl Viologen Radicals. Quantitative Estimation of Stored Electrons. *The Journal of Physical Chemistry C* 111, 9012-9015.

Liu, X., Ding, Z., He, Y., Xue, Z., Zhao, X., Lu, X., 2010. Electrochemical Behavior of Hydroquinone at Multi-Walled Carbon Nanotubes and Ionic Liquid Composite Film Modified Electrode. *Colloids and Surfaces B: Biointerfaces* 79, 27-32.

Monge-Romero, I.C., Suárez-Herrera, M.F., 2013. Electrocatalysis of the Hydroquinone/Benzoquinone Redox Couple at Platinum Electrodes Covered by a Thin Film of Poly(3,4-Ethylenedioxythiophene). *Synthetic Metals* 175, 36-41.

Mumme, J., Srocke, F., Heeg, K., Werner, M., 2014. Use of Biochars in Anaerobic Digestion. *Bioresource Technology* 164, 189-197.

Nurmi, J.T., Tratnyek, P.G., 2002. Electrochemical Properties of Natural Organic Matter (NOM), Fractions of NOM, and Model Biogeochemical Electron Shuttles. *Environmental Science & Technology* 36, 617-624.

Orlović-Leko, P., Vidović, K., Plavšić, M., Ciglencečki, I., Šimunić, I., Minkina, T., 2016. Voltammetry as a Tool for Rough and Rapid Characterization of Dissolved Organic Matter in the Drainage Water of Hydroameliorated Agricultural Areas in Croatia. *Journal of Solid State Electrochemistry* 20, 3097-3105.

Quan, M., Sanchez, D., Wasylkiw, M.F., Smith, D.K., 2007. Voltammetry of Quinones in Unbuffered Aqueous Solution: Reassessing the Roles of Proton Transfer and Hydrogen Bonding in the Aqueous Electrochemistry of Quinones. *Journal of the American Chemical Society* 129, 12847-12856.

Schneider, M.P.W., Hilf, M., Vogt, U.F., Schmidt, M.W.I., 2010. The Benzene Polycarboxylic Acid (BPCA) Pattern of Wood Pyrolyzed between 200°C and 1000°C. *Organic Geochemistry* 41, 1082-1088.

Schneider, M.P.W., Pyle, L.A., Clark, K.L., Hockaday, W.C., Masiello, C.A., Schmidt, M.W.I., 2013. Toward a “Molecular Thermometer” to Estimate the Charring Temperature of Wildland Charcoals Derived from Different Biomass Sources. *Environmental Science & Technology* 47, 11490-11495.

Spokas, K.A., 2010. Review of the Stability of Biochar in Soils: Predictability of O:C Molar Ratios. *Carbon Management* 1, 289-303.

Vanek, S.J., Lehmann, J., 2015. Phosphorus Availability to Beans Via Interactions between Mycorrhizas and Biochar. *Plant and Soil* 395, 105-123.

White, G.F., Shi, Z., Shi, L., Wang, Z., Dohnalkova, A.C., Marshall, M.J., Fredrickson, J.K., Zachara, J.M., Butt, J.N., Richardson, D.J., Clarke, T.A., 2013. Rapid Electron Exchange

between Surface-Exposed Bacterial Cytochromes and Fe(III) Minerals. *Proceedings of the National Academy of Sciences* 110, 6346-6351.

Wotton, B.M., Gould, J.S., McCaw, W.L., Cheney, N.P., Taylor, S.W., 2012. Flame Temperature and Residence Time of Fires in Dry Eucalypt Forest. *International Journal of Wildland Fire* 21, 270-281.

Xu, J., Chen, Q., Swain, G.M., 1998. Anthraquinonedisulfonate Electrochemistry: A Comparison of Glassy Carbon, Hydrogenated Glassy Carbon, Highly Oriented Pyrolytic Graphite, and Diamond Electrodes. *Analytical Chemistry* 70, 3146-3154.

Xu, W., Pignatello, J.J., Mitch, W.A., 2013. Role of Black Carbon Electrical Conductivity in Mediating Hexahydro-1,3,5-Trinitro-1,3,5-Triazine (RDX) Transformation on Carbon Surfaces by Sulfides. *Environmental Science & Technology* 47, 7129-7136.

Xu, W., Pignatello, J.J., Mitch, W.A., 2015. Reduction of Nitroaromatics Sorbed to Black Carbon by Direct Reaction with Sorbed Sulfides. *Environmental Science & Technology* 49, 3419-3426.

Yuan, Y., Yuan, T., Wang, D., Tang, J., Zhou, S., 2013. Sewage Sludge Biochar as an Efficient Catalyst for Oxygen Reduction Reaction in an Microbial Fuel Cell. *Bioresource Technology* 144, 115-120.

Zachara, J.M., Kukkadapu, R.K., Fredrickson, J.K., Gorby, Y.A., Smith, S.C., 2002. Biomineralization of Poorly Crystalline Fe(III) Oxides by Dissimilatory Metal Reducing Bacteria (DMRB). *Geomicrobiology Journal* 19, 179-207.

Reviewers' Comments:

Reviewer #1 (Remarks to the Author)

Re-review of manuscript "Rapid electron transfer by the carbon matrix in natural pyrogenic carbon"

by Sun and coworkers

Submitted to Nature Communications

General comments

This is a re-review of a revised manuscript that was originally submitted to Nature Communications earlier this year. The authors of the manuscript provided a detailed point-by-point response to the reviewer's comments (reviewer 1 in the attached document). The reviewer feels that the responses and the changes in the manuscript address the comments. The reviewer also appreciates that the authors took efforts to conduct additional experiments and that the results from these experiments provided additional evidence in support of the authors arguments.

The reviewer re-read the revised manuscript and feels that it has improved in clarity and quality compared to the original submission (which already was at a very high level). The reviewer has identified some points, mostly minor, that still need to be addressed. Overall, however, the reviewer recommends that the manuscript is accepted after minor revisions.

Specific comments

Abstract

Line 7: should the authors not directly refer to surface redox-active functional groups? Also, not all redox active groups are reversible (e.g. mono-hydroxylated are irreversibly oxidized)

Line 14: "1.5 V potential range for reduction" ... The wide potential range certainly is relevant also to processes other than mineral reduction. Not clear why this process needs to be emphasized in the abstract.

"Introduction"

line 25: "may mediate" instead of „is able to mediate"

Line 65: "maximum kinetic" ... please rephrase. There is no such thing as a "kinetic"

Line 116: Please consider rephrasing "probably due to the less homogenous contact". (i) mediated chronoamperometry does not require contact of the analyte (here: pyrogenic carbon) with the working electrode as electron transfer occurs via the mediator. The current wording is misleading in that one may think contact is required. (ii) also, it seems as if a contact cannot be more or less "homogenous". The authors should simply state that only a small fraction of the redox-active moieties of the materials were in contact with the working electrode that allowed for electron transfer

Line 127: "lower control" ... consider rephrasing for the same reasons. It seems that control cannot be "low". Maybe impaired pH buffering?

Line 154: "exposed to electron transfer". The reviewer is not sure what the authors mean here. The surfaces over which electron transfer is believed to occur? Please rephrase. A surface can be exposed to solution, air ... or possibly even to electrons. But not to electron transfer (also because the surface is part of this process).

Lines 189-191: Could it be that the author see metal-catalyzed H⁺ reduction at these potentials? Would this not be more logical than formation of Fe(0). Did the authors attempt to measure the pH dependency of the "Fe(II) reduction"? If a -60mV/pH dependency is observed, then it seems this would be indicative of H⁺ reduction to H₂ rather than reduction of Fe(II) to Fe(0).

Line 216: "During an increase" please rephrase. Maybe better: resulting from an increase ... or caused by an increase

Line 246: (Duration and pressure) ... what about O₂ availability? The reviewer would think that this must be a key factor. Consider adding.

Line 258: The authors should be aware (and possibly mention) that these studies were conducted in DMSO. As far as the reviewer can remember, CVs were featureless in water. So citing this paper is a bit misleading in the context of the discussion if the authors do not acknowledge that an organic solvent had to be used to identify reduction peaks. Also, in the paper by Nurmi and Tratnyek, it was unclear how potentials measured in DMSO relate to potentials measured in water

Line 283: Did the authors attempt to estimate the average degree of ring condensation for the low temperature chars? For instance from calculated aromaticity indices? The reviewer wonders because if the authors can show that the average degree of condensation is around unity, then using benzoquinone/hydrobenzoquinone as a model seems justified. This may well be true for the low-T chars. If, however, the average degree of condensation is higher, then this would raise the question if naphtho- or even anthraquinones are the better models.

Methods

Line 316: why was 1 hour instead of 30 min chosen for the 650°C char. The authors should provide an explanation and discuss potential effects of the longer dwell time on the properties of the 650°C char relative to the other chars in the T series

Page 28, Figure 4: Should it read MnO₂ in the panel b) inset instead of MnO? Also, the other minerals are hematite and magnetite not hematit and magnetit.

Reviewer #2 (Remarks to the Author)

The authors have worked hard to address both sets of reviewers comments and have produced important new evidence about the nature of the quinoa groups as well as better defining the source of their wood source. I think that the conclusions of the work are properly justified and that this paper will be well cited in the future.

Responses to reviewers' comments

We are grateful to the reviewers for their appreciation of our responses and the first revision of the manuscript. Below is the point-by-point response to the reviewers' remaining concerns. Upon the editor's request, all texts in supplementary information have now been moved to either the results or methods section of the main manuscript. All changes are highlighted by change-track function.

Sincerely,
Tianran Sun (on behalf of authors)

Reviewers' comments:

Reviewer #1

General comments:

This is a re-review of a revised manuscript that was originally submitted to Nature Communications earlier this year. The authors of the manuscript provided a detailed point-by-point response to the reviewer's comments (reviewer 1 in the attached document). The reviewer feels that the responses and the changes in the manuscript address the comments. The reviewer also appreciates that the authors took efforts to conduct additional experiments and that the results from these experiments provided additional evidence in support of the authors arguments.

The reviewer re-read the revised manuscript and feels that it has improved in clarity and quality compared to the original submission (which already was at a very high level). The reviewer has identified some points, mostly minor, that still need to be addressed. Overall, however, the reviewer recommends that the manuscript is accepted after minor revisions.

Author response: We greatly appreciate the reviewer's continued effort in improving the manuscript, and we are glad that our first revision addressed the reviewer's major concerns. All minor revisions suggested by the reviewer have been implemented in this second revision to further improve the clarity of the manuscript.

Specific comments:

Abstract

Line 7: should the authors not directly refer to surface redox-active functional groups? Also, not all redox active groups are reversible (e.g. mono-hydroxylated are irreversibly oxidized).

Author response: We agree with the reviewer. This sentence has been removed, and now reads "Surface functional groups constitute major electroactive components in pyrogenic carbon."

Line 14: “1.5 V potential range for reduction” ... The wide potential range certainly is relevant also to processes other than mineral reduction. Not clear why this process needs to be emphasized in the abstract.

Author response: The reason we emphasized the 1.5 V potential range for mineral reduction is to highlight the biogeochemical relevance of the electron transfer by pyrogenic carbon matrices. The reviewer is absolutely right that this potential range is relevant to many other processes than mineral reduction, so we revised this sentence to “have a 1.5 V potential range for biogeochemical reactions that invoke electron transfer processes”.

“Introduction”

line 25: “may mediate” instead of “is able to mediate”.

Author response: This sentence has been revised accordingly.

Line 65: “maximum kinetic” ... please rephrase. There is no such thing as a “kinetic”.

Author response: We meant “kinetics”, it has been revised. Thanks the reviewer for pointing that out.

Line 116: Please consider rephrasing “probably due to the less homogenous contact”. (i) mediated chronoamperometry does not require contact of the analyte (here: pyrogenic carbon) with the working electrode as electron transfer occurs via the mediator. The current wording is misleading in that one may think contact is required. (ii) also, it seems as if a contact cannot be more or less “homogenous”. The authors should simply state that only a small fraction of the redox-active moieties of the materials were in contact with the working electrode that allowed for electron transfer.

Author response: We agree with the reviewer that the current expression is ambiguous as we had not intended to state that the mediated chronoamperometry requires direct contact of analyte and electrode. This sentence has been rephrased to what the reviewer suggested.

Line 127: “lower control” ... consider rephrasing for the same reasons. It seems that control cannot be “low”. Maybe impaired pH buffering?

Author response: We agree with the reviewer and this sentence has been revised to “impaired pH buffering on pyrogenic carbon”.

Line 154: “exposed to electron transfer”. The reviewer is not sure what the authors mean here. The surfaces over which electron transfer is believed to occur? Please rephrase. A surface can be exposed to solution, air ... or possibly even to electrons. But not to electron transfer (also because the surface is part of this process).

Author response: We agree with the reviewer and this sentence has been revised to “carbon surfaces through which the electron transfer occurs”.

Lines 189-191: Could it be that the author see metal-catalyzed H⁺ reduction at these potentials? Would this not be more logical than formation of Fe(0). Did the authors attempt to measure the pH dependency of the “Fe(II) reduction”? If a -60mV/pH dependency is observed, then it seems this would be indicative of H⁺ reduction to H₂ rather than reduction of Fe(II) to Fe(0).

Author response: We agree with the reviewer that the Fe(II) catalyzed H⁺ reduction is more thermodynamically favorable than Fe(II) reduction to Fe(0) and have revised this discussion accordingly. We did not measure the pH dependency of Fe(II) reduction since the point of this figure is to demonstrate the broad potential range of the pyrogenic carbon matrices instead of Fe(II) reduction.

Line 216: “During an increase” please rephrase. Maybe better: resulting from an increase ... or caused by an increase.

Author response: This sentence has been revised to “caused by an increase”.

Line 246: (Duration and pressure) ... what about O₂ availability? The reviewer would think that this must be a key factor. Consider adding.

Author response: O₂ availability has been added.

Line 258: The authors should be aware (and possibly mention) that these studies were conducted in DMSO. As far as the reviewer can remember, CVs were featureless in water. So citing this paper is a bit misleading in the context of the discussion if the authors do not acknowledge that an organic solvent had to be used to identify reduction peaks. Also, in the paper by Nurmi and Tratnyek, it was unclear how potentials measured in DMSO relate to potentials measured in water.

Author response: We thank the reviewer for reminding us of the organic solvent use in the literature we cited. We have revised this discussion by acknowledging this point.

Line 283: Did the authors attempt to estimate the average degree of ring condensation for the low temperature chars? For instance from calculated aromaticity indices? The reviewer wonders because if the authors can show that the average degree of condensation is around unity, then using benzoquinone/hydrobenzoquinone as a model seems justified. This may well be true for the low-T chars. If, however, the average degree of condensation is higher, then this would raise the question if naphtho- or even anthraquinones are the better models.

Author response: We performed the aromaticity index (AI) calculations on our pyrogenic carbon samples and obtained the AI values close to unity at 0.8 to 1 from low-T to high-T pyrogenic carbon, which further supported our application of benzo- and hydroquinone as model compounds. AI was calculated based on the elemental (C, H, O) composition (see Supplementary Table 1) of pyrogenic carbon (Koch and Dittmar, 2006).

Methods

Line 316: why was 1 hour instead of 30 min chosen for the 650°C char. The authors should

provide an explanation and discuss potential effects of the longer dwell time on the properties of the 650°C char relative to the other chars in the T series.

Author response: Pyrogenic carbon produced at 650°C is in the transition region of slow to fast electron transfers of the studied carbon matrices (see Figure 1e). The redox peaks appeared very noisy and unstable when 30 min dwell time was applied. Therefore, we applied a slightly longer dwell time (1 hour) to improve the stability of the electron transfer by pyrogenic carbon produced at 650°C. The influence of a longer dwell time on the calculation of rate constants is minor and only account for less than 1% of overpotential decrease and less than 2% of ln(current) increase, which is still in the range of the standard deviation of replicate tests. This discussion has been added to the Method section.

Page 28, Figure 4: Should it read MnO₂ in the panel b) inset instead of MnO? Also, the other minerals are hematite and magnetite not hematit and magnetit.

Author response: We thank the reviewer for pointing out these typos. This figure has been revised accordingly.

Reviewer #2

The authors have worked hard to address both sets of reviewers comments and have produced important new evidence about the nature of the quinoa groups as well as better defining the source of their wood source. I think that the conclusions of the work are properly justified and that this paper will be well cited in the future.

Author response: We greatly appreciate the reviewer's effort in improving the manuscript, and we are glad that our revisions have addressed the reviewer's concerns.

Reference cited:

Koch, B.P., Dittmar, T., 2006. From Mass to Structure: An Aromaticity Index for High-Resolution Mass Data of Natural Organic Matter. *Rapid Communications in Mass Spectrometry* 20, 926-932.